# Broadly neutralizing anti-HIV-1 antibodies tether viral particles at the surface of infected cells

Jérémy Dufloo [1,2,8], Cyril Planchais[3], Stéphane Frémont[4], Valérie Lorin [3], Florence Guivel-Benhassine[1], Karl Stefic [5], Nicoletta Casartelli[1], Arnaud Echard [4], Philippe Roingeard [6], Hugo Mouquet[3], Olivier Schwartz [1,7✉] & Timothée Bruel [1,7✉]

Broadly neutralizing antibodies (bNAbs) targeting the HIV-1 envelope glycoprotein (Env) are promising molecules for therapeutic or prophylactic interventions. Beyond neutralization, bNAbs exert Fc-dependent functions including antibody-dependent cellular cytotoxicity and activation of the complement. Here, we show that a subset of bNAbs targeting the CD4 binding site and the V1/V2 or V3 loops inhibit viral release from infected cells. We combined immunofluorescence, scanning electron microscopy, transmission electron microscopy and immunogold staining to reveal that some bNAbs form large aggregates of virions at the surface of infected cells. This activity correlates with the capacity of bNAbs to bind to Env at the cell surface and to neutralize cell-free viral particles. We further show that antibody bivalency is required for viral retention, and that aggregated virions are neutralized. We have thus identified an additional antiviral activity of bNAbs, which block HIV-1 release by tethering viral particles at the surface of infected cells.

[1] Institut Pasteur, Université de Paris, CNRS UMR3569, Virus and Immunity Unit, 75015 Paris, France. [2] Université de Paris, École doctorale BioSPC 562, 75013 Paris, France. [3] Institut Pasteur, Université de Paris, INSERM U1222, Humoral Immunology Laboratory, 75015 Paris, France. [4] Institut Pasteur, Université de Paris, CNRS UMR3691, Membrane Traffic and Cell Division Unit, 75015 Paris, France. [5] CHRU de Tours, Hôpital Bretonneau, Service de Bactériologie-Virologie, 37000 Tours, France. [6] Université de Tours, CHRU de Tours, INSERM U1259 MAVIVH and Plateforme IBiSA de Microscopie Électronique, 37000 Tours, France. [7] Vaccine Research Institute, 94000 Créteil, France. [8] Present address: Institute for Integrative Systems Biology (I2SysBio), Universitat de València-CSIC, 46980 València, Spain. ✉email: olivier.schwartz@pasteur.fr; timothee.bruel@pasteur.fr

The HIV-1 envelope (Env) glycoprotein is a trimer of gp41/gp120 heterodimers. It is the only viral protein present at the surface of viral particles. Env mediates entry into target cells, which makes it the target of neutralizing antibodies. Broadly neutralizing antibodies (bNAbs) have been isolated from patients called "elite neutralizers" and inhibit the majority of HIV-1 strains[1]. They target conserved sites of vulnerability at the surface of Env: the CD4 binding site (CD4bs), the N-glycans associated with the V1/V2 and V3 loops, the silent face of gp120, the membrane proximal external region (MPER) of gp41 and a larger site spanning the interface between gp41 and gp120. In both non-human primates and humanized mice, infusion of bNAbs decreases viral loads[2,3], prevents infection[4–6] and delays viral rebound[7,8]. Several bNAbs are under clinical evaluation[9,10]. The anti-CD4bs bNAbs 3BNC117 and VRC01, and the anti-V3 10–1074 decrease viral loads in HIV-1-infected viremic individuals[11–13]. 3BNC117, alone or in combination with 10–1074, delays viral rebound after antiretroviral therapy (ART) interruption[14,15].

In addition to neutralizing viral particles, bNAbs recognize infected cells and recruit immune effectors through their Fc domain. Such Fc-dependent activities include antibody-dependent cellular cytotoxicity (ADCC)[16–18], antibody-dependent phagocytosis (ADCP)[19] or complement activation[20]. The contribution of Fc-effector functions to antibody-mediated protection against simian-human immunodeficiency virus (SHIV) acquisition in animal models is debated[21–24]. The discrepant results may depend on the capacity of the tested antibody to neutralize the virus used for challenge, with the contribution of Fc-effector functions increasing as neutralization potency decreases. Fc-effector functions boost bNAbs therapeutic efficacy in macaques[25] and are required to efficiently target the reservoir in humanized mice[7]. Therefore, the non-neutralizing activities of bNAbs contribute to their in vivo efficacy.

A few antibodies directed against other viruses inhibit the assembly or the release of viral particles. For instance, a neutralizing monoclonal antibody (mAb) directed against Chikungunya virus (CHIKV) blocks envelope-driven viral assembly, leading to the intracellular accumulation of immature viral-like particles and inhibition of viral release[26,27]. A mAb against influenza virus inhibits viral egress by extracellularly aggregating mature viral particles[28]. Similar inhibition of release was demonstrated for antibodies targeting Marburg (MARV) or Ebola (EBOV) viruses[29,30]. The impact of anti-HIV-1 bNAbs on viral release has yet to be examined.

The steps of HIV-1 budding are well characterized. HIV-1 Gag p55 are anchored in the inner leaflet of the plasma membrane, forming a lattice that initiates particle formation[31]. In parallel, Env accumulates at the budding site through interactions between the gp41 cytoplasmic tail and Gag[31]. This is followed by ESCRT-mediated budding, viral release and particle maturation by the viral protease, which cleaves Gag into virion-associated proteins p24, p17, p7, p6, p2 and p1[32]. Whether bNAbs interfere with these processes is unknown.

Here, we show that a subset of bNAbs inhibits HIV-1 release from infected CD4 T cells. bNAbs tether mature viral particles as large extracellular immune complexes without inhibiting budding or maturation.

## Results

### bNAbs inhibit HIV-1 release from infected CD4 T cells.
We first asked whether bNAbs impair HIV-1 release. To this aim, we infected primary CD4 T cells with HIV-1 for two days, washed the cells to replace the medium and then subjected infected cells to treatment with a panel of bNAbs or an isotype control (mGO53) for 24 h. Since the capacity of bNAbs to neutralize HIV-1 varies across antibodies and viral strains, we added antiretrovirals (azidothymidine [AZT] and lamivudine [3TC]) during bNAbs treatment (Fig. 1a). To further avoid any confounding effect of viral replication, we treated cells with antibodies at the time of peak viral replication (Supplementary Fig. 1a). This strategy allowed the normalization of the frequency of infected cells across isotype- and bNAb-treated conditions (Supplementary Fig. 1a). AZT and 3TC are reverse-transcriptase inhibitors that act early in the viral cycle, without interfering with p24 production by cells productively infected at the time of addition. Accordingly, p24 production is reduced but not halted by addition of AZT-3TC (Supplementary Fig. 1b). We analyzed viral release in supernatants by ELISA and assessed cell-associated Gag by flow cytometry and microscopy (Fig. 1a). We used three HIV-1 strains: the lab-adapted AD8 isolate, a transmitted/founder strain (CH058) and a clade B virus isolated from the reservoir of an ART-treated patient (vKB18)[16]. We first used two bNAbs, 10–1074 and 3BNC117, which target the V3 loop and the CD4bs, respectively. We normalized results to the condition without antibody. Both bNAbs, but not the isotype control, decreased p24 levels in the supernatant (Fig. 1b). This reduction reached 64% with 10–1074 in vKB18-infected cells. This effect was not due to a residual inhibition of viral spread by bNAbs, as the frequency of infected cells (as measured by a Gag-specific staining) was similar regardless of the antibody tested (Supplementary Fig. 1c). We then analyzed infected cells (defined as CD4$^-$ Gag$^+$) by flow cytometry (Supplementary Fig. 1d). Both 10–1074 and 3BNC117 increased the median fluorescence intensity (MFI) of cell-associated Gag (Fig. 1c, d). The most potent effect was again observed with 10–1074 in vKB18-infected cells (1.8-fold increase in Gag MFI compared to control antibody). A kinetic analysis performed with 10–1074 and CH058-infected cells revealed that Gag MFI increases over 24 h (Supplementary Fig. 1e). Of note, adding extra PBS washes or a proteolytic degradation of cell surface proteins by trypsin prior to bNAbs treatment had no impact on the increase in MFI induced by 10–1074 (Supplementary Fig. 2a). We then tested a panel of 17 anti-Env antibodies including bNAbs and non-neutralizing antibodies (nnAbs) (Fig. 1e, Supplementary Fig. 2b, c, and Supplementary Table 1). For the three viral strains, bNAbs targeting the V3 and V1/V2 loops or the CD4bs were the most efficient at decreasing viral release and increasing cell-associated Gag. Both activities were correlated and dose-dependent (Supplementary Figs. 2d, e and 3). In contrast, bNAbs targeting the MPER, the gp120/gp41 interface and nnAbs were inactive.

We next examined the binding of bNAbs to infected cells. All antibodies recognized cells infected with the three viral strains, but to different extents (Fig. 1e and Supplementary Fig. 4a–c). As previously observed, the most potent binders were anti-V3 loop bNAbs[16,20]. Binding of antibodies to infected cells strongly correlated with their ability to increase cell-associated Gag and to inhibit HIV-1 release (Fig. 1f and Supplementary Fig. 3). Antibodies inhibiting viral egress displayed a higher MFI of binding to infected cells than those that did not (Supplementary Fig. 4d). Consistently, sera from 16 ART-treated patients (at a 1:100 dilution) did not reach the levels of binding of the most potent bNAbs and were unable to increase Gag levels in CH058-infected cells (Supplementary Fig. 4e).

Overall, these results suggest that bNAbs capable of highly potent binding at the plasma membrane decrease viral release by retaining viral material in infected cells.

### bNAbs retain mature viral particles at the plasma membrane.
To determine whether bNAbs trigger the accumulation of Gag p55 precursor or its virion-associated cleavage products (e.g., p24, p17 and p7), we examined infected cells by western blot. We selected the CH058 strain and 10–1074, 3BNC117, N6, PGT128, PGDM1400 and 10E8 for this analysis. 10–1074 increased the

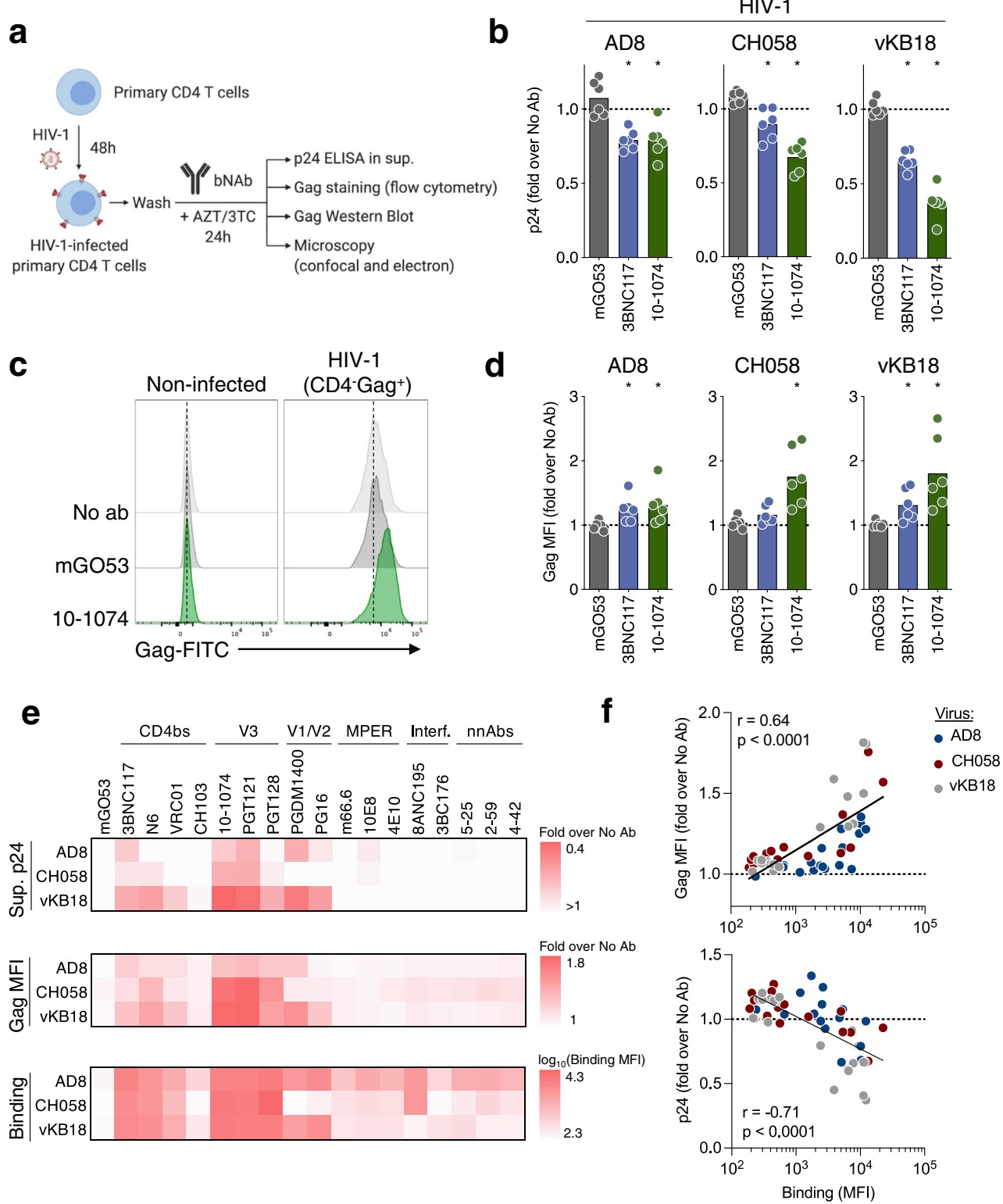

levels of p24 (3.6-fold compared to the isotype control) (Fig. 2a, b and Supplementary Fig. 5a). A similar increase was observed with 3BNC117 (1.3-fold), N6 (2.2-fold), and PGT128 (2.2-fold) (Supplementary Fig. 5b). Of note, 10–1074 and other antibodies slightly but significantly decreased p55 levels (Fig. 2a, b and Supplementary Fig. 5b). PGDM1400 and 10E8, which did not increase cell-associated Gag by flow cytometry with this strain, did not promote p24 accumulation (Supplementary Fig. 5b). To

determine whether Gag accumulation was associated with similar changes of viral RNA, we performed a RT-qPCR assay on CH058-infected cells. 10–1074 increased the amount of genomic unspliced (US) viral RNA associated with virions, but not the levels of multiply spliced (MS) forms generated during replication (Supplementary Fig. 5c).

We next determined by confocal microscopy the localization of Gag in CH058-infected cells exposed to 10–1074. The analysis of

**Fig. 1 bNAbs inhibit HIV-1 release. a** Schematic of the experimental strategy. Primary CD4 T cells were infected with HIV-1 for 48 h, washed, and cultivated for 24 h with antibodies (anti-HIV-1 Env or isotype control; 100 nM, corresponding to 15 μg/mL) in the presence of azidothymidine (AZT) and lamivudine (3TC) prior to subsequent analyses. **b** p24 levels in the supernatants of CD4 T cells infected with three strains of HIV-1 (AD8, CH058 or vKB18) and cultivated for 24 h with an isotype control (mGO53) or bNAbs (3BNC117 and 10–1074). p24 concentrations were measured by ELISA and normalized to the "no antibody" condition. Each dot represents a donor of CD4 T cells ($n = 6$). *$p = 0.0313$ (two-tailed Wilcoxon test compared to mGO53). Bars represent the mean. **c** Representative flow cytometry histograms of Gag staining in non-infected (left) and infected (HIV-1 strain CH058 (CD4⁻Gag⁺); right) cells after 24 h of culture without antibody (No ab), with an isotype control (mGO53) or with a bNAb (10–1074). The dotted lines indicate the median fluorescence intensity (MFI) of the "No ab" condition. **d** Cell-associated Gag levels of CD4 T cells infected with different strains of HIV-1 and cultivated for 24 h with an isotype control or bNAbs. The MFI of staining was measured by flow cytometry and normalized to the "no antibody" condition. Each dot represents a donor of CD4 T cells ($n = 6$). *$p = 0.0313$ (two-tailed Wilcoxon test compared to mGO53). Bars represent the mean. **e** Fold-change in the p24 levels in the supernatants (top) and the cell-associated Gag levels (middle) were measured in CD4 T cells infected with three strains of HIV-1 (AD8, CH058, and vKB18) and cultivated with non-neutralizing antibodies (nnAbs) or bNAbs targeting various epitopes of the envelope (CD4 binding site [CD4bs], V3 loop, V1/V2 loop, Membrane Proximal External Region [MPER] and gp120/gp41 interface). Levels of binding to infected cells for the same antibodies are also depicted (bottom). Data represent the mean of 6 donors of CD4 T cells. **f** Correlation between the fold-change in cell-associated Gag, the fold-change in supernatant p24 levels and antibody binding to infected cells. Each dot represents a different antibody ($n = 18$). Mean values of six donors of CD4 T cells are depicted. Data from cells infected with three different viral strains are represented with different colors. A two-tailed Spearman correlation test was performed, and the correlation $r$ and $p$-value are indicated. Antibodies were tested at a concentration of 100 nM, corresponding to 15 μg/mL. Source data are provided as a Source Data file.

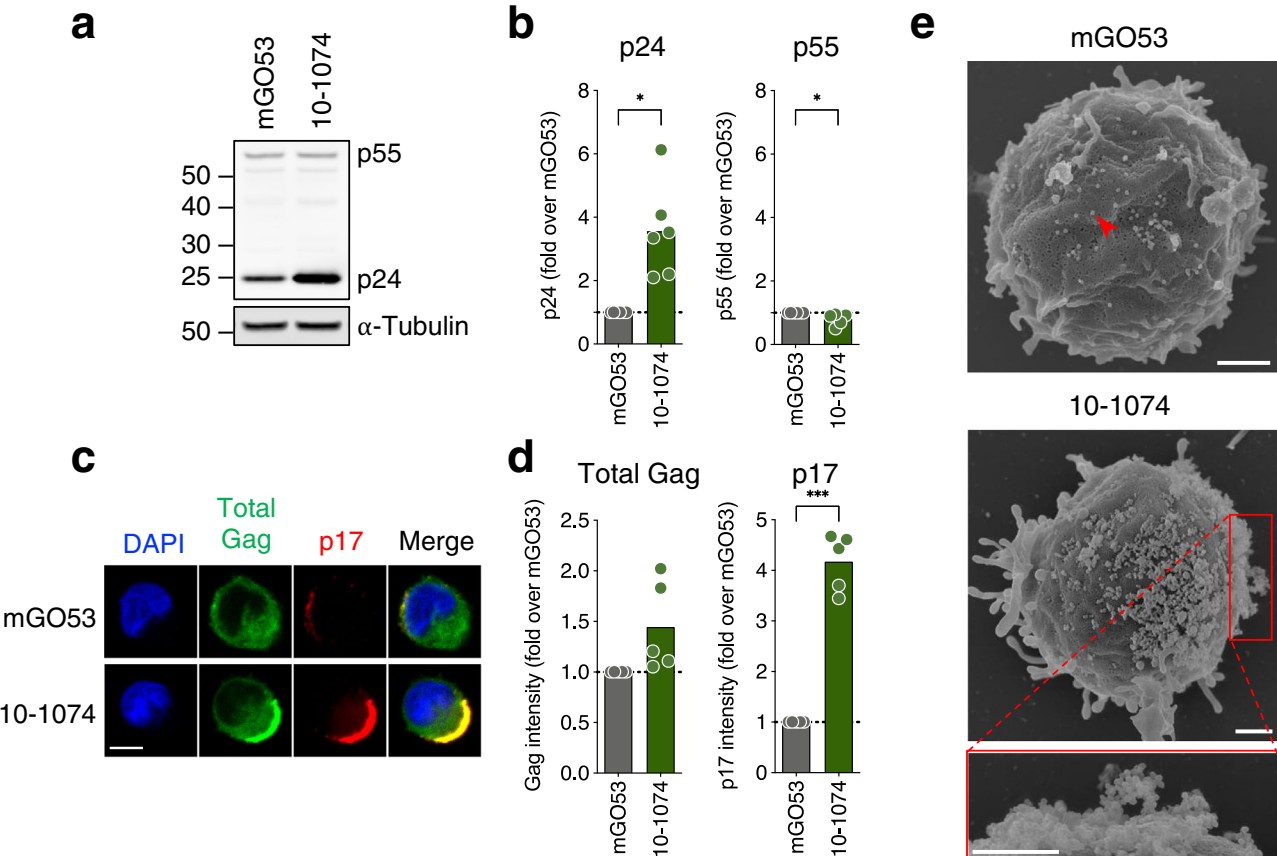

**Fig. 2 bNAbs retain mature HIV-1 particles at the plasma membrane. a** Western blot analysis of Gag in CD4 T cells infected with HIV-1 (CH058) for 48 h and then cultivated 24 h with an isotype control (mGO53) or a bNAb (10–1074). α-Tubulin was used as a loading control. Molecular weights (kDa) are indicated on the left. One representative experiment (out of six) is shown. **b** Western blot analysis of p24 (left) and p55 (right) levels in HIV-1-infected (CH058) CD4 T cells cultivated for 24 h with an isotype control (mGO53) or a bNAb (10–1074). Results are normalized to α-Tubulin and to mGO53. Each dot represents a donor of CD4 T cells ($n = 6$). *$p = 0.0313$ (two-tailed Wilcoxon test). Bars represent the mean. **c** Representative confocal microscopy images of infected CD4 T cells (CH058) cultivated for 24 h with an isotype control (mGO53) or with a bNAb (10–1074). Cells were stained for intracellular total Gag (green) and p17 (red). Nuclei were stained with DAPI (blue). Scale bar: 5 μm. One representative experiment (out of five) is shown. **d** Confocal microscopy analysis of total Gag and p17 intensity levels in infected (CH058) CD4 T cells cultivated for 24 h with an isotype control (mGO53) or with a bNAb (10–1074). Results are expressed as the fold-change in Gag staining intensity over mGO53. Each dot represents a donor of CD4 T cells ($n = 5$). At least 50 cells were analyzed per donor. ***$p = 0.0002$ (two-tailed paired t-test). Bars represent the mean. **e** Representative scanning electron microscopy images of infected (CH058) CD4 T cells cultivated for 24 h with an isotype control (mGO53) or with a bNAb (10–1074). A red arrowhead points an HIV-1 viral particle as an example. Scale bar: 1 μm. One representative experiment (out of two) is shown. Antibodies were tested at a concentration of 100 nM, corresponding to 15 μg/mL. Source data are provided as a Source Data file.

803 cells across 5 donors of CD4 T cells revealed that this bNAb induced a strong accumulation of p17 at the plasma membrane (4.2-fold increase compared to mGO53) (Fig. 2c, d and Supplementary Fig. 5d, e). As increased p17 levels may be due to capture of extracellular virions, we also examined 150 non-infected bystander cells from these 5 donors. This analysis revealed a lack of increase of p17 on bystander cells (Supplementary Fig. 5f), showing that p17 accumulation by 10–1074 requires productive infection. To gain better insight into these peri-membranous p17$^+$ aggregates, we performed correlative light and scanning electron microscopy of CH058-infected cells. 10–1074 induced massive extracellular aggregates of viral particles, sometimes forming atypical bridge-like 3D structures (Fig. 2e and Supplementary Fig. 6).

Altogether, the data show that bNAbs act at a late stage of the viral life cycle, by retaining mature viral particles at the plasma membrane.

**bNAbs retain viral particles as large immune complexes at the plasma membrane.** We then investigated whether the Fc region and antibody bivalency are required for inhibition of viral release. To this aim, we generated monovalent Fab and bivalent F(ab')$_2$ fragments of 10–1074 and compared their activity to full-length IgG. Antibodies and their fragments were tested at a concentration of 100 nM. mGO53 Fab and F(ab')$_2$ fragments were included as negative controls (Supplementary Fig. 7). 10–1074 Fab and F(ab')$_2$ similarly bound to CH058-infected cells (Fig. 3a). 10–1074 F(ab')$_2$ inhibited viral release and increased cell-associated Gag levels, whereas the Fab fragment was inactive (Fig. 3a). Similar results were obtained with the AD8 and vKB18 strains (Supplementary Fig. 7). Of note, the 10–1074 F(ab')$_2$ bound to infected cells less efficiently than the full-length antibody and was thus slightly less active at retaining virions at the cell surface (Supplementary Fig. 7). Altogether, our data show that bivalency, but not the Fc region, is required to trap viral particles.

To further characterize the inhibition of viral egress, we analyzed the ultrastructure of viral aggregates and the localization of bNAbs by transmission and scanning electron microscopy, coupled to immunogold staining of 10–1074 in CH058-infected cells. We observed large extracellular membrane-tethered immune complexes of virions tangled with 10–1074. (Fig. 3b–e and Supplementary Figs. 8 and 9). Trapped virions make contacts with budding viral particles (Fig. 3b and Supplementary Fig. 10). We further observed a bNAb immunogold staining in between a budding virus and a tethered virion (Fig. 3d). Antibodies were detected at the surface of individual virions as well as in regions where two or more virions were in close contact. The antibodies were very rarely detected at the plasma membrane, suggesting that they bound mostly to the Env complexes present on virions rather than at the cell surface. No intracellular accumulation of viral products or budding arrest was observed in the presence of antibodies. Similar results were obtained in CH058-infected CHME microglial macrophage-like cells cultured in the presence of 10–1074, suggesting that the phenotype is not restricted to CD4 T cells (Supplementary Fig. 11).

Overall, our data support a model in which bNAbs trigger the formation of aggregates of viral particles, after their budding, a process that further prevents their egress into the supernatant.

**Aggregated virions are neutralized.** We then determined the stability of aggregates and asked whether tethering of viral particles is an additional process to the antiviral activity of bNAbs. First, we measured viral aggregation during three days in the presence or absence of antiretrovirals (Fig. 4a and Supplementary Fig. 12a). Aggregation peaked at day 1 and then decreased,

regardless of the presence of antiretrovirals. A similar profile was observed when performing a p17 staining (Supplementary Fig. 12b). To further determine the half-life of viral aggregates, we generated viral aggregates by treating CH058-infected cells with 10–1074 for 24 h, washed them to remove bNAbs, and followed the fate of cell-associated aggregates by flow cytometry. Viral aggregation was readily measured at day 0 and undetectable 24 h after the wash (Supplementary Fig. 12c). Altogether these results suggest that viral aggregation at the surface of infected cells is transient, with a half-life below 24 h.

Next, we asked whether viral aggregates are neutralized. We measured the cell-free neutralizing activity of our panel of antibodies against the three HIV-1 strains (Fig. 4b). As expected, most bNAbs efficiently neutralized these viruses. The two primary strains, CH058 and vKB18 were less sensitive to neutralization. CH058 was not neutralized by V1/V2-targeting bNAbs (PGDM1400 and PG16) while vKB18 resisted those targeting the MPER and the gp120/gp41 interface. The capacity of bNAbs to tether viral particles (as measured by the decrease of p24 release) strongly correlated with their neutralization IC$_{50}$ (Fig. 4c and Supplementary Fig. 12d). We then assessed the infectivity of infected cells and supernatants over the course of a three-day bNAb treatment in the absence of AZT/3TC. Infected cells were treated with an HIV-1 protease inhibitor (Ritonavir) to disentangle the infectivity of already formed cell-associated viral aggregates and newly produced viral particles. Values were normalized to a condition without antibody to determine a relative infectivity. Neither cells nor supernatants were infectious after addition of bNAbs, while isotype-treated cells and their supernatants were infectious (Fig. 4d). This result shows that infected cells harboring viral aggregates are neutralized by bNAbs and that released viral particles lost their infectious potential.

We then determined the impact of retention on the overall antiviral activity of bNAbs. To this aim, we compared the activity of 10–1074 IgG and its Fab, in the presence or absence of antiretrovirals, and measured viral replication by quantifying p24 in the supernatant. The full-length IgG mediates both retention and neutralization whereas the Fab only mediates neutralization. In the absence of antiretrovirals, the full-length IgG decreased p24 release by 30% as soon as day 1, up to the end of the follow up at day 3 (Fig. 4e). The Fab induced a significant decrease in p24 production only at day 2, reaching the levels of the full antibody at day 3. In the absence of viral replication (i.e., with antiretrovirals), the full antibody displayed antiviral activity, whereas the Fab was inactive (Fig. 4e). Altogether, these data show that the full-length IgG targets infected cells to decrease p24 independently of its neutralization activity. Retention is thus an early antiviral activity of bNAbs, which allows a rapid decrease of p24 release, prior to further viral inhibition by neutralization.

Altogether, these results show that viral aggregation and neutralization are additive processes.

## Discussion

Here, we show that some bNAbs inhibit viral release by tethering mature viral particles at the cell membrane. The antiviral activities of bNAbs go well beyond neutralization[33]. bNAbs trigger ADCC, complement and phagocytosis of either infected cells or viral particles, all requiring the Fc-domain and interaction with cognate Fc receptors on immune effector systems. We report that F(ab')$_2$ fragments lacking the Fc domain inhibit viral egress, albeit slightly less efficiently than full-length bNAbs. This difference is probably due to either a lower binding on infected cells or to Fc-Fc interactions[34], which may facilitate the retention of viral particles by stabilizing immune complexes. Thus, the viral retention by bNAbs reported here may be considered as a Fc-independent antiviral activity of HIV-1 antibodies.

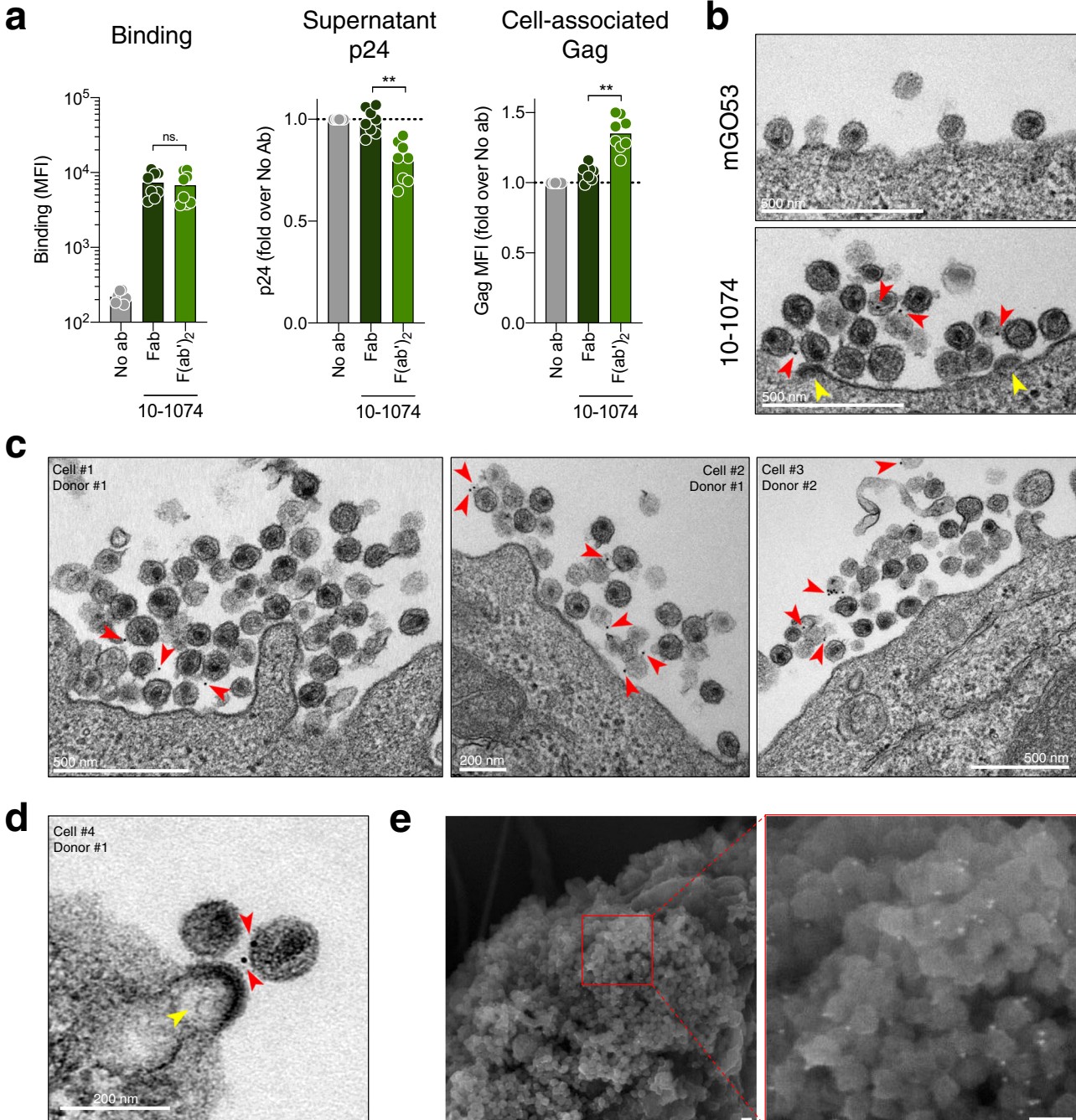

**Fig. 3 Tethering of viral particles by bNAbs requires antibody bivalency. a** Primary CD4 T cells were infected with HIV-1 (strain CH058) for 48 h and then were cultivated for 24 h with monovalent Fab and bivalent F(ab')$_2$ fragments of the bNAb 10–1074. Binding of the antibody fragments (left), and fold-change in released p24 (middle) and cell-associated Gag (right) were measured. Each dot represents a donor of CD4 T cells ($n = 8$). **$p = 0.0078$, ns. not significant (two-tailed Wilcoxon test). Bars represent the mean. Antibodies were tested at a concentration of 100 nM, **b** Infected CD4 T cells (CH058) cultivated for 24 h with an isotype control (mGO53) or with a bNAb (10–1074) were stained with an anti-human IgG antibody coupled to colloidal gold beads and analyzed by transmission electron microscopy (TEM). Representative images are shown. Red arrowheads point colloidal gold beads, indicative of 10–1074. Yellow arrowheads indicate budding viral particles. Scale bar: 500 nm. One representative experiment (out of two) is shown. **c** Other examples of 10–1074-induced retention observed with TEM. The cell donor and the cell ID is indicated for each image. Scale bar: 200 or 500 nm. **d** Example of viral retention at budding site, including a bNAb immunogold staining in between a viral particle and a budding virion. One representative experiment (out of two) is shown. **e** Infected CD4 T cells (CH058) cultivated for 24 h with a bNAb (10–1074) were stained with an anti-human IgG antibody coupled to colloidal gold and analyzed by scanning electron microscopy. A representative image is shown. Bright white dots are the colloidal gold beads, indicative of 10–1074. A magnification on an area with strong colloidal gold staining is shown. Scale bar: 100 nm. One representative experiment (out of two) is shown. Source data are provided as a Source Data file.

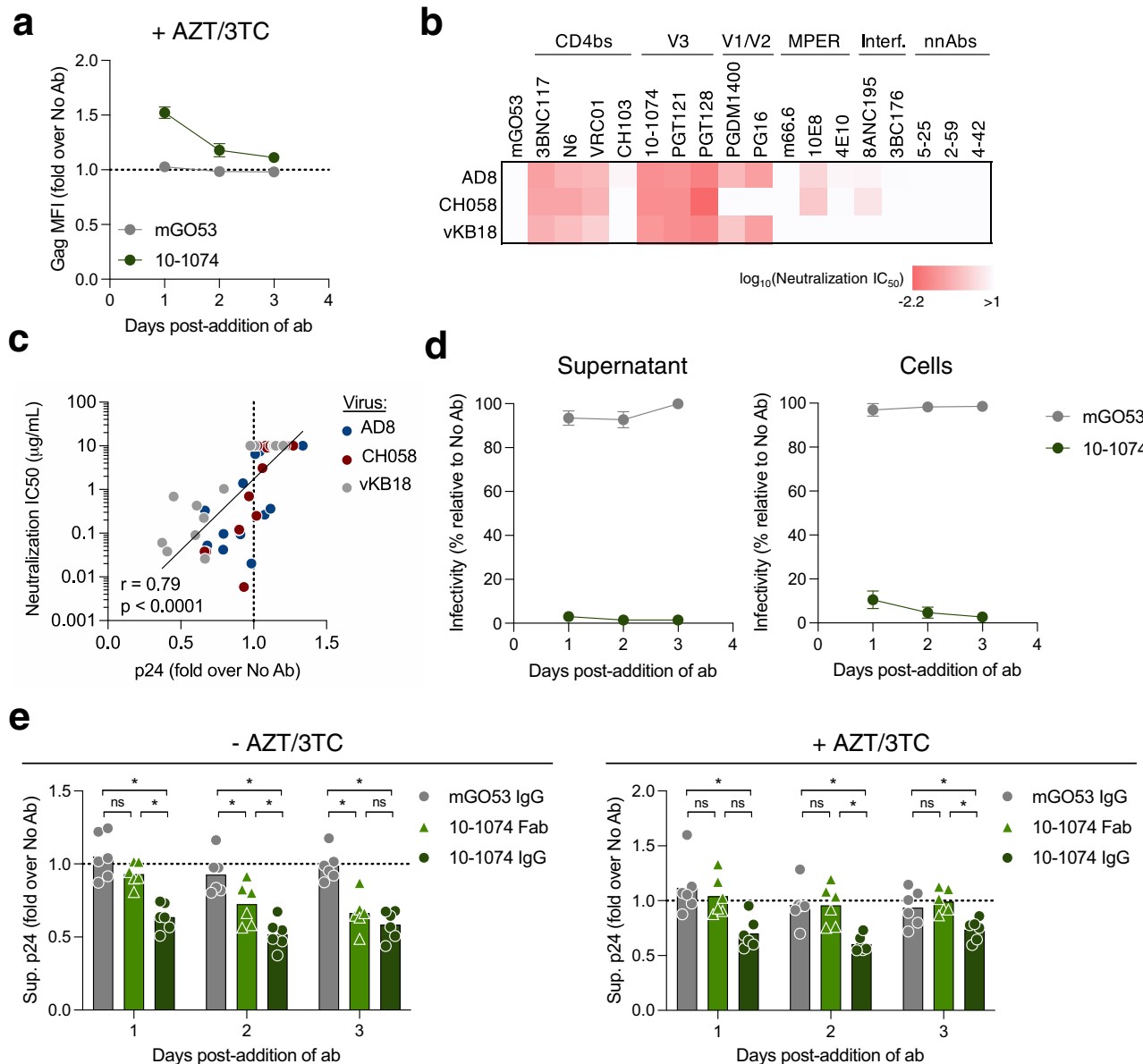

**Fig. 4 bNAbs-mediated aggregates are neutralized. a** Primary CD4 T cells were infected with HIV-1 for 48 h and then cultivated with a bNAb (10–1074) or an isotype control (mGO53) in the presence of AZT/3TC. Fold-change in cell-associated Gag was measured by flow cytometry after 1, 2, and 3 days of culture. Values are normalized to the "no antibody" condition. Data represent the mean ± SEM of 6 donors of CD4 T cells. **b** Neutralization inhibitory concentrations 50% ($IC_{50}$) of non-neutralizing antibodies (nnAbs) or bNAbs targeting various epitopes of the envelope (CD4-binding site [CD4bs], V3 loop, V1/V2 loop, membrane proximal external region [MPER] and gp120/gp41 interface) against the three strains of HIV-1 (AD8, CH058 and vKB18). Data represent the mean of two independent experiments. **c** Correlation between the fold-change in supernatant p24 levels and antibody $IC_{50}$ of neutralization. Each dot represents a different antibody ($n = 18$). Mean values of six donors of CD4 T cells are depicted for p24 fold-changes. Data from cells infected with three different viral strains are represented with different colors. A two-tailed Spearman correlation test was performed, and the correlation $r$ and $p$-value are indicated. **d** Primary CD4 T cells were infected with HIV-1 for 48 h and then cultivated for 24 h with 10–1074 in the absence of AZT/3TC to form aggregates. The infectivity of supernatants (left) and cells (right) was measured on TZM-bl cells. Data represent the mean ± SEM of 6 (left) and 3 (right) donors of CD4 T cells. **e** Primary CD4 T cells were infected with HIV-1 for 48 h and then cultivated for 24 h with 10–1074 or its monovalent Fab fragment, without (left) or with (right) AZT/3TC. p24 concentrations were measured by ELISA and normalized to the "no antibody" condition. Each dot represents a donor of CD4 T cells ($n = 6$). *$p = 0.0313$ (Two-tailed Wilcoxon test compared to mGO53). Bars represent the mean. Antibodies and Fab fragments were tested at a concentration of 100 nM. Source data are provided as a Source Data file.

This activity was observed with eight out of fourteen bNAbs and none of the three nnAb tested. The most potent antibodies were those binding with a high efficacy to infected cells, with a high capacity to neutralize cell-free viral particles and targeting the CD4bs and V3 regions of Env. There were strong correlations between the blocking activity, the levels of antibody binding and neutralization $IC_{50}$. This may explain the lack of activity of nnAbs

and polyclonal antibodies from infected individuals, as they recognized infected cells less efficiently than bNAbs[35–37]. However, among the 17 bNAbs and the three strains tested here, only a few combinations of antibodies and viral isolates reached a reduction of p24 release above 50%. It remains to be determined how this may contribute to their overall antiviral efficacy. Furthermore, we tested antibodies at 15 µg/mL (100 nM). This is not

necessarily the optimal binding concentration, which may vary depending on the antibody. We cannot exclude that viral aggregation would be more efficient for some antibodies at higher concentrations. Experiments with a larger panel of mAb and purified IgG from patients will help determining whether other Env domains may allow antibody-mediated viral retention, including non-neutralizing epitopes. Unexpectedly, some antibodies that strongly bound to infected cells poorly tethered viral particles. This indicates that binding alone is not sufficient to retain viral particles, and that aggregation of different virions necessitates the attachment of antibodies to selective regions of Env. For instance, PGT128 is unable to trap viral particles, while targeting the same N332 glycan patch as 10–1074. Thus, subtle differences in epitopes influence viral retention by antibodies. Indeed, antibody-mediated tethering by antibodies likely requires efficient binding to Env anchored in both viral and in cellular membranes, as well as the capacity to bridge two virions, which may not be possible for all antibodies or epitopes. It will be also important to determine the nature and composition of viral aggregates triggered by bNAbs to determine whether cellular proteins or components of the extracellular matrix contribute to their formation.

The phenomenon reported here is reminiscent of the effect of Tetherin/Bst-2, an interferon-induced cellular restriction factor that inhibits viral egress by trapping viral particles at the cell surface[38,39]. Tetherin is counteracted by the viral accessory protein Vpu[38,39]. Viral release is thus a vulnerable step of HIV-1 replication, which is targeted by both innate and humoral immunity. Interestingly, Tetherin senses viral aggregates and activates the NF-κB signaling pathway[40]. Env/bNAbs complexes at the surface of infected cells can also end up being internalized[41]. Future work will help determining whether the bNAb-induced viral aggregates enhance, or on the contrary, down-modulate HIV-1 innate sensing. We observed that viral aggregates disappear from the surface of infected cells in about a day. It will be worth assessing their fate, and examining whether they are internalized or if they detach from the cells surface. The capacity of macrophages or other antigen-presenting cells to capture these aggregates and subsequently impact adaptive and inflammatory immune responses would also deserve investigation.

Our transmission electron microscopy data indicate that bNAb-mediated inhibition of HIV-1 release was not associated with an obvious block of viral budding. However, the possibility remains that bNAbs or the aggregates themselves indirectly impede budding. Our data strongly suggest that bNAbs retain virions at budding sites. The sites of aggregation observed by transmission electron microscopy contained budding viral particles. We did not observe aggregates on bystander non-infected cells, confirming that productive infection is needed to form aggregates. We have previously reported that bNAbs decrease viral capture by CD4 T target cells[36,42]. The mature viral particles present within aggregates may still have been trapped immediately after budding and maturation, or may represent released virions that have been captured by these large structures. These questions may be addressed by measuring the impact of bNAbs on the kinetics of viral budding and aggregation by real-time high-resolution imaging[43]. Formation of immune complexes at the budding site has been observed for different viral species, including IAV, MARV and EBOV, and here HIV-1. Whether this non-neutralizing activity is a general feature of antibodies targeting other enveloped viruses will deserve further investigation.

We observed an impairment of viral budding by bNAbs in primary CD4 T cells and CHME macrophage-like cells. In myeloid cells, viral budding occurs in intracellular compartments and escapes antibody recognition[44–46]. It will be of interest to characterize how bNAbs may inhibit release in primary myeloid cells as well as the potential immunological consequences of this effect. It will also be of importance to determine whether inhibition of viral release by bNAbs occurs in vivo. Our cell culture experiments do not recapitulate mechanical stresses that occur in vivo (e.g., blood flow, high cell density, tissue extracellular matrix, etc..), which may destabilize immune complexes formed between antibodies and viral particles. In animal models and in humans, the passive administration of bNAbs boosts host antiviral cellular and humoral immune responses[47–49], a phenomenon known as "the vaccinal effect" of antibodies. The underlying mechanisms are not fully understood. One hypothesis is that antigen-presenting cells uptake immune complexes formed by bNAbs[50]. The origin of such immune complexes is still not well understood. In cell-free systems, anti-HIV-1 IgG poorly mediate aggregation of virions but only at a narrow concentration range[51,52]. Here we show that some bNAbs create immune complexes directly from infected cells, likely because the high concentration of Env and viral particles at budding sites favors their formation[53]. Whether the viral aggregates formed at the surface of infected cells contribute to the vaccinal effect of antibodies will deserve future investigation.

In conclusion, our data reveal that a subset of anti-HIV-1 bNAbs inhibit viral release from infected cells and highlight the polyfunctionality of antibody-mediated antiviral responses.

## Methods

**Cells**. Peripheral blood mononuclear cells (PBMCs) were isolated from peripheral blood of healthy human donors from the Etablissement Français du Sang, in accordance with local ethical guidelines. CD4 T cells were obtained from PBMCs by positive immunomagnetic selection (Miltenyi) and activated for 3 days in the presence of Interleukin 2 (IL-2; 50 IU/mL; R&D Systems) and phytohemagglutinin (PHA; 1 μg/mL; Oxoid). CD4 T cells were cultivated in Roswell Park Memorial Institute (RPMI) medium (Gibco) supplemented with 10% fetal calf serum (FCS; Gibco) and 1% penicillin/streptomycin (PS; Gibco).

293T cells were obtained from the ATCC (CRL3216™) and tested negative for mycoplasma (Mycoplasma Detection Kit; Lonza). They were cultivated in Dulbecco's Modified Eagle Medium (DMEM; Gibco) supplemented with 10% FCS and 1% PS.

**Viruses and infections**. HIV-1 strains NLAD8 (lab-adapted) and CH058 (transmitted-founder; obtained from the NIH AIDS Reagent Program) were prepared by the transfection of 293T cells along with vesicular stomatitis virus G to normalize infectivity. vKB18 was isolated from the reservoir of an ART-treated HIV-1-infected individual with undetectable viral load[16] and amplified on activated CD4 T cells from healthy individuals. No more than two passages were performed. Cells were infected for 4 h at 37 °C with 30 s agitation every 10 min at a high concentration ($10 \times 10^6$ cells/mL) to maximize virus/cells contacts in the presence of HEPES (10 mM; Gibco) and DEAE/Dextran (16 μg/mL). Cells were then cultivated at $1 \times 10^6$ cells/mL in the presence of IL-2 (50 IU/mL; R&D Systems) for 2 days before performing further experiments. Of note, viral release was inhibited by 10–1074 irrespectively of the presence of DEAE/Dextran during the initial infection of T cells.

**Antibodies and antibody fragments**. Anti-Env monoclonal antibodies and isotypic control mGO53 were produced as recombinant human IgG1 monoclonal antibodies by co-transfection of Freestyle 293-F suspension cells (Thermo Fisher Scientific) using the PEI precipitation method as previously described[54]. Anti-Env antibodies' variable domains were also cloned into human Fab- and F(ab')₂-expressing vectors as previously described[55] and produced as aforementioned. Recombinant monoclonal IgG1 antibodies, and corresponding Fab and F(ab')₂ fragments were purified by batch/gravity-flow affinity chromatography using respectively protein G sepharose 4 fast flow beads (GE Healthcare) and Ni-sepharose Excel beads (GE Healthcare) following the manufacturers' procedures.

**Human sera**. Sera from ART-treated HIV-infected patients were used as a polyclonal source of anti-Env antibodies (dilution 1:100). Each participant provided written consent to participate in the study, which was approved by the regional investigational review board (IRB; Comité de Protection des Personnes Ile-de-France VII, Paris, France) and performed according to the European guidelines and the Declaration of Helsinki. Sera from healthy donors were used as controls. Patients' characteristics are presented in Supplementary Table 2.

**Culture of infected cells with bNAbs.** After washing out viral inoculum, HIV-1-infected primary CD4 T cells were cultivated for 24 h at $1 \times 10^6$ cells/mL in presence of IL-2 (50 IU/mL; R&D Systems), azidothymidine (AZT; 10 µM; Sigma) and lamivudine (3TC; 2.5 µM; NIH AIDS Reagent Program). Anti-HIV-1 antibodies or an isotype control (mGO53) were added to the culture (100 nM, unless otherwise stated). After 24 h, supernatants and cells were processed for further analysis (see below).

**p24 ELISA.** Supernatants were inactivated with 1% Triton X-100. 96 well plates were washed with PBS and coated with an anti-p24 antibody (Nittobo, Cat#HIV-018–48304; 5 µg/mL) overnight at 37 °C. Plates were saturated with PBS 10% FCS for 2 h at 37 °C. Samples were incubated for 2 h at 37 °C. An in-house-biotinylated detection anti-p24 antibody (clone 36/5.4A; Zeptometrix; 0.5 µg/mL) was added for 1 h at 37 °C followed by Streptavidin-conjugated Horse Radish Peroxydase (HRP; BD Pharmingen; dilution 1:3000) for 1 h at 37 °C. O-phenylenediamine dihydrochloride (OPD; Thermo Fisher Scientific) solution was added for 30 min at room temperature in the dark and the reaction was stopped with $H_2SO_4$ 2 M. Optical density at 492 nm was measured. A standard curve was obtained using dilutions of a supernatant with known p24 content and used to infer absolute p24 concentrations. Results were normalized to the condition without antibody.

**Flow cytometry analysis of cell-associated Gag levels.** Cells were washed and incubated with a PE-conjugated anti-CD4 antibody (clone VIT4; Miltenyi; dilution 1:200) for 30 min at 4 °C. Cells were fixed with 4% PFA for 10 min at room temperature. Cells were washed and stained with a FITC-conjugated anti-Gag antibody (clone KC57; Beckman-Coulter; diluted 1:500 in PBS/BSA 1%/Saponin 0.05%) for 30 min at room temperature. Cells were washed and resuspended in PBS. Data were acquired on an Attune NxT flow cytometer (Life Technologies) using the Attune NxT Software (Life Technologies). Data were analyzed using the FlowJo software (v10.7; BD). The Median Fluorescence Intensity (MFI) of Gag in infected (CD4$^-$Gag$^+$) cells was calculated and normalized to the condition without antibody.

**Western blotting.** Cells were washed with PBS and lysed for 30 min on ice using PBS Triton X-100 1% containing protease inhibitors (Roche). Cell lysates were cleared by centrifugation at $17,000 \times g$ for 10 min at 4 °C. Protein concentration was measured using the Protein Assay Dye Reagent Concentrate (Bio-Rad). Heat-denaturated (10 min at 72 °C) lysates (20–30 µg) were separated by sodium dodecyl sulfate-polyacrylamide gel electrophoresis (SDS-PAGE) and transferred onto a nitrocellulose membrane using the iBlot 2 Dry Blotting System (Invitrogen). The following primary antibodies were used: mouse monoclonal anti-α-Tubulin (clone AA13; Sigma-Aldrich; diluted 1:5000), rabbit polyclonal anti-p24 (NIH AIDS Reagent Program; diluted 1:1000). The following fluorescently labeled secondary antibodies were used: DyLight 680-conjugated Goat anti-Mouse IgG (Invitrogen; dilution 1:5000), DyLight 800-conjugated Goat anti-Rabbit IgG (Invitrogen; dilution 1:5000). Fluorescent signals were detected with a LI-COR Odyssey scanner and analyzed using the ImageStudioLite software.

**Immunofluorescence.** Coverslips were coated with Poly-L-Lysine 0.01% for 10 min at room temperature and dried for 2 h. Cells were incubated for 30 min at 37 °C to let them adhere to the coverslip. Medium was removed and cells were fixed with PFA 4% for 10 min. Cells were incubated for 1 h with the following primary antibodies diluted in PBS/BSA 1%/Saponin 0.05%: mouse IgG1 monoclonal anti-Gag (clone KC57; Beckman-Coulter; dilution 1:75) and mouse IgG2a monoclonal anti-p17 (NIBSC CFAR; Cat#0342; dilution 1:100). After three washes, cells were incubated for 1 h with the following secondary antibodies diluted in PBS/BSA 1%/Saponin 0.05%: Alexa Fluor 488-conjugated rabbit anti-FITC (Invitrogen; dilution 1:100) and Cy3-conjugated goat anti-mouse IgG2a (Jackson ImmunoResearch; dilution 1:500). Coverslips were washed with PBS and water and mounted in DAPI-containing mounting medium (SouthernBiotech). Images were acquired on a LSM 700 confocal microscope (Zeiss) using ZEN software (ZEISS). Images were analyzed with Fiji software. Briefly, regions of interests (ROI) were drawn manually around each infected cell. Intensities of total Gag-Alexa Fluor 488 and p17-Cy3 signals were calculated in each ROI. Between 49 and 138 cells were analyzed per condition.

**Binding of bNAbs to infected cells.** $0.5–1 \times 10^5$ HIV-1-infected cells were incubated 30 min at 37 °C with anti-Env antibodies or with an isotype human IgG1 control (mGO53) at 100 nM. Cells were washed and incubated 30 min at 4 °C with an Alexa Fluor 647-conjugated anti-human IgG1 (H + L) antibody (Life Technologies; 1:400 dilution). Cells were then fixed with 4% PFA and stained for intracellular Gag (see above). Data were acquired on an Attune Nxt instrument (Life Technologies) and analyzed using FlowJo software.

**Scanning electron microscopy (SEM).** Glass bottom dishes with a gridded coverslip (MatTek) were coated with Poly-L-Lysine 0.01% for 10 min at room temperature and dried overnight. For correlative Light-SEM, cells were incubated in the dishes for 30 min at 37 °C in presence of 3BNC117 (100 nM). Cells were washed and incubated with a secondary Alexa Fluor 647-conjugated Goat anti-

Human IgG (H + L) (Invitrogen; dilution 1:400) for 30 min at 4 °C. For Immunogold-SEM, cells were stained for 30 min at 4 °C with an anti-human IgG antibody coupled to 12 nm colloidal gold beads (Jackson ImmunoResearch; dilution 1:20 in PBS/BSA/EDTA) prior to incubation in the dishes (30 min at 37 °C). Cells were then fixed for 1 h in PHEM (60 mM PIPES, pH 6.9; 25 mM HEPES; 2 mM $MgCl_2$; 10 mM EGTA) containing 2.5% of Glutaraldehyde and 0.5% of PFA. Cells were then washed three times with PHEM. For correlative Light-SEM, cells were imaged using an inverted Eclipse Ti-E microscope (Nikon) equipped with a CSU-X1 spinning disk confocal scanning unit (MDS), driving a EMCCD Camera (Evolve 512 Delta, Photometrics) to identify and locate infected cells and their location on the grid. Cells were then post-fixed with 2% osmium tetroxide ($OsO_4$) in 1 M HEPES buffer at room temperature for a maximum of 1 h, and washed with distilled water. The samples were dehydrated by transferring into successive baths of ethanol (10, 25, 50, 75 and 95%, 5 min each, and two times 100%, 10 min each) and dried into the critical point dryer's chamber (Leica EM CDP300). Then, samples were coated with 8 nm of gold/palladium (high-resolution ion beam coater, Gatan Model 681) or 7 nm of carbon for Immunogold-SEM experiments. The images were acquired with a JEOL JSM 6700 F field emission scanning electron microscope.

**Transmission electron microscopy.** Cells were incubated with an anti-human IgG antibody coupled to 12 nm colloidal gold beads (Jackson ImmunoResearch; dilution 1:20 in PBS/BSA/EDTA) for 30 min at 4 °C. Cells were fixed for 24 h in 4% PFA and 1% glutaraldehyde (Sigma) in 0.1 M phosphate buffer (pH 7.2). Cells were washed in PBS and post-fixed with 2% osmium tetroxide (Agar Scientific) for 1 h. Cells were fully dehydrated in a graded series of ethanol solutions and propylene oxide. The impregnation step was performed with a mixture of (1:1) propylene oxide/Epon resin (Sigma) and left overnight in pure resin. Cells were then embedded in resin blocks, which were allowed to polymerize for 48 h at 60 °C. Ultra-thin sections (70 nm) of blocks were obtained with a Leica EM UC7 ultra-microtome (Wetzlar). Sections were stained with 5% uranyl acetate (Agar Scientific) and 5% lead citrate (Sigma), and observations were made with a JEOL 1011 transmission electron microscope.

**Quantitative real-time PCR for quantification of HIV-1 RNA.** Total RNA was extracted using the RNeasy Mini Kit (Qiagen). RNA concentration was measured with a NanoDrop instrument (Thermo Fisher Scientific). 400 ng of RNA were used for cDNA synthesis using SuperScript II reverse-transcriptase (Life Technologies). Unspliced (US) and multiply spliced (MS) HIV-1 were quantified with a semi-nested real-time PCR (qRT-PCR)[56]. The first amplification was performed using the following primers: US-forward: 5′-AAC TAG GGA ACC CAC TGC TTA AG-3′; US-reverse: 5′-TCT CCT TCT AGC CTC CGC TAG TC-3′; MS-forward: 5′-CTT AGG CAT CTC CTA TGG CAG GAA-3′; MS-reverse: 5′-TCA AGC GGT GGT AGC TGA AGA GG-3′. The run conditions were: 2 min at 94 °C and 16 amplification cycles of 94 °C for 30 s, 55 °C for 30 s and 72 °C for 30 s on a SimpliAmp thermocycler (Applied Biosystems). Two microliters of these reactions were used for qPCR quantification using iTaq Universal SYBR Green Supermix (Biorad) and the following primers: US-forward: 5′-TCT CTA GCA GTG GCG CCC GAA CA-3′; US-reverse: 5′-TCT CCT TCT AGC CTC CGC TAG TC-3′; MS-forward: 5′-CTT AGG CAT CTC CTA TGG CAG GAA-3′; MS-reverse: 5′-TTC CTT CGG GCC TGT CGG GTC CC-3′. The qPCR conditions were: 15 min at 95 °C and 40 amplification cycles of 95 °C for 15 s, 60 °C for 20 sec and 72 °C for 30 s on a Mastercycler® RealPlex[2] (Eppendorf). Levels of cellular transcripts were normalized to GAPDH. Results are expressed as fold induction relative to the isotype condition (mGO53) using the $2^{-\Delta\Delta Ct}$ method.

**TZM-bl neutralization and infectivity assays.** $1.5 \times 10^4$ TZM-bl were plated one day prior to the neutralization and infectivity assays. For neutralization, AD8, CH058 and vKB18 HIV-1 strains were incubated with anti-Env monoclonal antibodies[57–70] at the indicated concentrations for 1 h at room temperature and added to TZM-bl cells. Viral infectivity was determined by incubating $2 \times 10^4$ cells or supernatants (undiluted) with TZM-bl cells. In the case of co-culture with infected cells, an HIV-1 protease inhibitor was added (Ritonavir; 10 µM). Luciferase activity was measured 48 h later using an EnSpire plate reader (PerkinElmer). The percentage of neutralization was calculated using the following formula: $100 \times (1 - (\text{value with mAb} - \text{value in "non-infected"})/(\text{value in "no Ab"} - \text{value in "non-infected"}))$. Neutralizing activity of each monoclonal antibody was expressed as the $IC_{50}$ in µg/mL using a reconstructed curve with the percentage of neutralization at the different indicated concentrations. Infectivity was calculated using the following formula: $(\text{value with mAb} - \text{value in "non-infected"})/(\text{value in "no Ab"} - \text{value in "non-infected"})$.

**Data processing and statistical analysis.** Calculations were performed and figures were drawn using Excel 365 (Microsoft) or Prism 9.0 (Graphpad Software). Statistical analyses were performed using Prism 9.0 (GraphPad Software). Information about the statistical tests performed can be found in the figure legends.

**Reporting summary**. Further information on research design is available in the Nature Research Reporting Summary linked to this article.

## Data availability
All data supporting the findings of this study are available within the Article. Source data are provided with this paper.

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

## Acknowledgements

We thank members of the Virus and Immunity Unit for discussion and help. We thank Maaran Michael Rajah for critical reading of the manuscript. We thank the NIH AIDS Reagent Program for providing reagents. We thank Fabienne Arcanger for the preparation of the TEM samples. We thank the imaging facility Ultrapole of the Institut Pasteur. We thank Céline Desouche and Damien Thierry from the French HIV National Reference Center for subtyping the vKB18 strain. O.S. is funded by Institut Pasteur, Agence Nationale de Recherches sur le Sida et les Hépatites Virales (ANRS), Sidaction, the Vaccine Research Institute (ANR-10-LABX-77), Labex IBEID (ANR-10-LABX-62-IBEID), "TIMTAMDEN" ANR-14-CE14-0029, "CHIKV-Viro-Immuno" ANR-14-CE14-0015-01, and the Gilead HIV cure program. J.D. is funded by a Ph.D grant from the French Ministry of Higher Education and Research. H.M. is funded by the Institut Pasteur, the Milieu Intérieur Program (ANR-10-LABX-69-01), INSERM, ANRS, and Gilead HIV cure program. C.P. was supported by an ANRS fellowship. A.E. and S.F. are funded by Institut Pasteur, CNRS, and ANRS (ANRS-21020 AP2020-2). P.R. is funded by INSERM, Université de Tours and ANRS.

## Author contributions

Conceptualization and methodology: J.D., O.S., T.B. Acquisition or analysis of data: J.D., C.P., S.F., V.L., F.G-B., K.S., N.C., P.R., H.M., O.S., T.B. Data assembly and manuscript writing: J.D., O.S., T.B. Funding acquisition: A.E., P.R., H.M., O.S. Supervision: O.S., T.B.

## Competing interests

The authors declare no competing interests.
