## [Peer Review File · Nature Communications]

REVIEWER COMMENTS

Reviewer #1 (Remarks to the Author):

Antiviral activity of antibodies by inhibiting virus release has been demonstrated in several viral diseases but not for HIV-1. Dufloo and colleagues close this gap by exploring the capacity of broadly neutralizing antibodies (bNAbs) to block virus release from infected cells. Indeed as the authors convincingly demonstrate, some bNAbs appear to have this capacity. The authors demonstrate the effect of bNAbs on infected cells by decreased p24 levels in the supernatant and the concomitant increase of cell-associated Gag levels (measured by flow cytometry and Western blot). Visualization of extracellular trapped virions, cross-linked by full antibodies or (Fab)₂ but not Fab by confocal and electron microscopic analysis, demonstrate that bivalent antibody molecules but not the Fc tail of Abs is required for generating antibody aggregates. The results are very intriguing but raise a number of questions that need to be addressed.

Specific points:

1. The authors do not address the relevance of their findings to antiviral activity. Are these bAb virus aggregates stable? Are the virions completely inactivated? Or do the aggregates dissociate over time and release infectious virus? Can the aggregated virions regain infectivity in cell to cell spread?
2. Overall the fraction of viruses that is trapped based on the GAG retention and p24 release is modest. The author should try to estimate the effect on virus production over multiple rounds of infection.
3. As shown in Figure S1 A, the percentage of Gag-specific cells stained by HIV-1-Abs was very low and hardly distinguishable from control Ab mG053. This needs to be discussed.
4. Line 125: "We next examined the binding of bNAbs to infected cells. All antibodies recognized cells infected with the three viral strains, but to different extents". This validation of binding capacity is an important piece of information and should be presented already with Figure 1.
5. Fold differences in p24 content and GAG MFI are in some Ab-virus combinations significantly different from untreated cells but the difference is in several cases very modest. In fact only few combinations reach a 50% difference. These are V3 glycan bNAbs against CH058 and vKB18 for both p24 and GAG measurements and V1/V2 bNAbs against vKB18 for p24. The authors need to discuss what impact such a low reduction in infectivity may have.
6. The differences between some bNAbs in the retention activity are not easy to comprehend. Why would 10-1074 and PGT128 which are highly related react differently? PGT128 is the best binder to CH058 and AD8 (Figure 3B) but has not effect on p24 release and is not the best in retention (Figure S2)?

7. In this context, it is often not entirely clear from the text whether an observed effect of an Ab is observed for tested strains or not. Better attention needs to be paid to the differential patterns. The correlation analysis in Figure 1F, 3C, and 3D should also be performed separately for the individual viruses, as it seems to be strongly driven by the vKB18 data.

8. Line 106: "PGDM1400 and 10E8, which did not increase cell-associated Gag by flow cytometry, did not promote p24 accumulation" . This statement is not entirely correct as it only reflects CH058 results. Data for isolates AD8 and vKB18 are different. Importantly, PGDM1400 did increase cell associated Gag for those strains by flow cytometry, Figure 1E).

9. Line 130: "nnAbs and sera from 16 ART treated patients (at a 1:100 dilution) did not reach the levels of binding of bnAbs and were unable to trap HIV-1 at the cell surface." This is not correct, AD8 binding is good for nnAbs according to Figure 3B.

10. Several antibodies show enhancing effects in the p24 release data (Figure S2) that are in the extent higher than the inhibitory effects seen with some bnAbs. This is significant of nnAb 4-42 for vKB18. By simple eye-balling it seems surprising that not more of these "enhancing" effects reach significance.

11. Neutralization data of all antibodies against the probed viruses should be shown and correlated to virus retention capacity.

12. The authors should discuss which of the probed Abs have the capacity of bivalent binding of two virions. This will sterically not be possible for all Abs and likely will lend some explanations to the observed patterns.

13. The discussion is very interesting to read as the authors layout the many open questions that remain to be addressed. Considering that there are many loose ends, short-comings of the current study should be noted. The fact that the impact of virus trapping on virus spread may be limited based on the present results, needs to be discussed.

14. "Glycans-V3 loops" is a rather unusual term.

15. Line 38: Since all other bnAb epitopes are mentioned, silent face bnAbs should be listed as well

16. Line 79: Why were the specific three virus strains chosen? Subtype and sensitivity to the probed Abs and sera should be provided.

17. Line 167: The authors propose "a Fc-independent non-neutralizing antiviral activity of HIV-1 antibodies". I am not sure that there is already enough data to state that this capacity is distinct from the neutralization activity. The virions in the aggregates may be neutralized, high affinity (as required for potent neutralization) may be the basis of efficient trapping

18. Figure S6B: Please indicate concentration of antibody used.

Reviewer #2 (Remarks to the Author):

Inhibition of viruses by formation of aggregates is a very classical phenomenon largely describe as the first mechanism participating in virus inhibition. In this manuscript, authors propose that this aggregation occurs by certain bNAbs during virus released on infected cells. This is an interesting additional mechanism of inhibition, but additional experiments should be added in order to proof that indeed, the virus detected is virus produced and released form cells and not the virus inoculum forming aggregates close to cell membrane.

In the experimental conditions described in this manuscript, author detect very high infection rates: from 2 to 15% infection for primary CD4 T cell (supplemental figure 1). However, in the result section, authors claimed that AZT and 3TC were used to “halt viral replication and avoid confounding effect due to neutralization” (line 82). This sentence is not clear, and misleading especially because later on, authors stipulate, “Ab inhibit virus released from infected cells” (line 23 and lane 160)? It would be extremely relevant to determine whether virus replicate or not under their experimental conditions. If the virus detected by cytometry and by ELISA correspond to virus produced after 24H (AZT and 3TC)+ 24H(AZT and 3TC + Ab) ie 48 H culture, why adding AZT +3TC. What would be this infection rate in the absence of inhibitors AZT+3TC? If AZT and 3TC is added at “suboptimal concentrations”, what is the rational of adding these inhibitors without stopping HIV infection? On contrary, if inhibitors (AZT and 3TC) were added to “halt viral replication”, the virus detected may correspond to the inoculum that forms immune complexes (virus/Ab) and bind directly on cells without infection (increased MFI of infected cells) and as a result, decreases the residual virus in the supernatant (p24 in ELISA). The conclusion would then be completely different. This accumulation of HIV on target cells by Abs may even be detrimental (enhancement at low Ab concentration!) for further HIV replication. Therefore, authors should demonstrate (by kinetic experiments, washing out virus inoculum after infection.... for example) that the virus detected on CD4 T cells effectively correspond to virus released from infected cells and not to residual inoculum trapped on cells via HIV specific Abs.

At the concentration used, the new bNAbs had almost no neutralizing activity on primary cells and viruses (see Cohen et al., J. Virol 2018 Feb 12;92(5):e01883-17. doi: 10.1128/JVI.01883-17. Print 2018 Mar 1.). At higher concentrations however, 10-1074 should neutralize infection. Therefore, the increased MFI detected in S2 D is very surprising. Again, what would be the inhibitory activity of 10-1074 at those concentrations in the absence of AZT + 3TC? Authors should analyze the neutralizing profile of the Abs added one hour after infection on primary cells using the same experimental conditions but in absence of AZT+3TC.

Additional comments

Why using the unique concentration of 100nM to compare the different mAbs knowing that the different HIV specific Ab used in this manuscript bind with different affinity on gp120, trimeric spike and viruses. This aggregation activity may be observed for all HIV specific Abs able to recognize HIV if used at optimal binding concentration. Please comment.

What is the rational of using DEA/Dextran. DEA/dextran will modify cell membrane. Does it influence virus capture on CD4 T cells? Does virus/Abs aggregate at the surface of cells without DEA/Dextran?

Reviewer #1 (Remarks to the Author):

Antiviral activity of antibodies by inhibiting virus release has been demonstrated in several viral diseases but not for HIV-1. Dufloo and colleagues close this gap by exploring the capacity of broadly neutralizing antibodies (bNAbs) to block virus release from infected cells. Indeed as the authors convincingly demonstrate, some bNAbs appear to have this capacity. The authors demonstrate the effect of bNAbs on infected cells by decreased p24 levels in the supernatant and the concomitant increase of cell-associated Gag levels (measured by flow cytometry and Western blot). Visualization of extracellular trapped virions, cross-linked by full antibodies or (Fab)2 but not Fab by confocal and electron microscopic analysis, demonstrate that bivalent antibody molecules but not the Fc tail of Abs is required for generating antibody aggregates. The results are very intriguing but raise a number of questions that need to be addressed.

We would like to thank reviewer #1 for his/her evaluation our work and his/her help to improve our study. We hope that the new version of our manuscript will answer his/her concerns.

Specific points:

1. The authors do not address the relevance of their findings to antiviral activity. Are these bnAb virus aggregates stable? Are the virions completely inactivated? Or do the aggregates dissociate over time and release infectious virus? Can the aggregated virions regain infectivity in cell to cell spread?

We agree that our study lacked a demonstration of the relevance of viral tethering by antibodies in the context of HIV-1. We have now analyzed the stability of viral aggregates over 3 days, in the presence or absence of viral replication, and the infectivity of viral particles and infected cells subjected to bNAbs-mediated aggregation. Altogether, these data show that viral aggregates are neutralized when attached to infected cells, but also upon release. We also compared the bNAb 10-1074 to its Fab (which lacks tethering capacity) to highlight that retention is cumulative with neutralization at early timepoints. These data are presented in a new Figure 4:

The results sections have been modified to include the description of this new figure. It reads (lines 172-212):

“We then determined the stability of aggregates and asked whether tethering of viral particles is a process additional to the antiviral activity of bNAbs. First, we measured viral aggregation during three days in the presence or absence of antiretrovirals (Figure 4A and Supplementary Figure 11A). Aggregation peaked at day 1 and then decreased, regardless of the presence of antiretrovirals. A similar profile was observed when performing a p17 staining (Supplementary Figure 11B). To further determine the half-life of viral aggregates, we generated viral aggregates by treating CH058-infected cells with 10-1074 for 24h, washed them to remove bNAbs, and followed the fate of cell-associated aggregates by flow cytometry. Viral aggregation was readily measured at day 0 and undetectable 24h after the wash (Supplementary Figure 11C). Altogether these results suggest that viral aggregation at the surface of infected cells is transient, with a half-life below 24h.

Next, we asked whether viral aggregates are neutralized. We measured the cell-free neutralizing activity of our panel of antibodies against the three HIV-1 strains (Figure 4B). As expected, most bNAbs efficiently neutralized these viruses. The two primary strains, CH058 and vKB18 were less sensitive to neutralization. CH058 was not neutralized by V1/V2-targeting bNAbs (PGDM1400 and PG16) while vKB18 resisted those targeting the MPER and the gp120/gp41 interface. The capacity of bNAbs to tether viral particles (as measured by the decrease of p24 release) strongly correlated with their neutralization IC_{50} (Figure 4C and Supplementary Figure 11D). We then assessed the infectivity of infected cells and supernatants over the course of a three-day bNAb treatment in the absence of AZT/3TC. Infected cells were treated with an HIV-1 protease inhibitor (Ritonavir) to disentangle the infectivity of already formed cell-associated viral aggregates and newly produced viral particles. Values were normalized to a condition without antibody to determine a relative infectivity. Neither cells nor supernatants were infectious after addition of bNAbs, while isotype-treated cells and their supernatants were infectious (Figure 4D). This result shows that infected cells harboring viral aggregates are neutralized by bNAbs and that released viral particles lost their infectious potential.

We then determined the impact of retention on the overall antiviral activity of bNAbs. To this aim, we compared the activity of 10-1074 IgG and its Fab, in the presence or absence of antiretrovirals, and measured viral replication by quantifying p24 in the supernatant. The full-length IgG mediates both retention and neutralization whereas the Fab only mediates neutralization. In the absence of antiretrovirals, the full-length IgG decreased p24 release by 30% as soon as day 1, up to the end of the follow up at day 3 (Figure 4E). The Fab induced a significant decrease in p24 production only at day 2, reaching the levels of the full antibody at day 3. In the absence of viral replication (i.e. with antiretrovirals), the full antibody displayed antiviral activity, whereas the Fab was inactive (Figure 4E). Altogether, these data show that the full-length IgG targets infected cells to decrease p24 independently of its neutralization activity. Retention is thus an early antiviral activity of bNAbs, which allows a rapid decrease of p24 release, prior to further viral inhibition by neutralization.

Altogether, these results show that viral aggregation and neutralization are additive processes.”

2. Overall the fraction of viruses that is trapped based on the GAG retention and p24 release is modest. The author should try to estimate the effect on virus production over multiple rounds of infection.

Indeed, the overall effect appears quite modest, and it is difficult to predict from our initial data how it may impact viral spread over multiple cycles. We have thus integrated in the new Figure 4 an experiment where we compared the Fab of 10-1074 (which is unable to form aggregates) to the full-length IgG. We tested their impact on the kinetics of viral spread over three days, in the presence or absence of

antiretroviral agents. This set of experiments reveals that viral aggregation acts in addition to neutralization at an early time point.

This is now stated in the results section as follows (lines 200-210):

“We then determined the impact of retention on the overall antiviral activity of bNAbs. To this aim, we compared the activity of 10-1074 IgG and its Fab, in the presence or absence of antiretrovirals, and measured viral replication by quantifying p24 in the supernatant. The full-length IgG mediates both retention and neutralization whereas the Fab only mediates neutralization. In the absence of antiretrovirals, the full-length IgG decreased p24 release by 30% as soon as day 1, up to the end of the follow up at day 3 (Figure 4E). The Fab induced a significant decrease in p24 production only at day 2, reaching the levels of the full antibody at day 3. In the absence of viral replication (i.e. with antiretrovirals), the full antibody displayed antiviral activity, whereas the Fab was inactive (Figure 4E). Altogether, these data show that the full-length IgG targets infected cells to decrease p24 independently of its neutralization activity. Retention is thus an early antiviral activity of bNAbs, which allows a rapid decrease of p24 release, prior to further viral inhibition by neutralization.”

3. As shown in Figure S1 A, the percentage of Gag-specific cells stained by HIV-1-Abs was very low and hardly distinguishable from control Ab mG053. This needs to be discussed.

We apologize for the lack of clarity. The aim of this figure is to confirm that antibody treatment does not affect the frequency of infected cells in our conditions. Thus, cells were stained with an anti-Gag antibody, explaining the lack of difference between mG053 and all other anti-Env antibodies.

We have rephrased the corresponding sentence in the results section to improve clarity (lines 92-94):

“This effect was not due to a residual inhibition of viral spread by bNAbs, as the frequency of infected cells (as measured by a Gag-specific staining) was similar regardless of the antibody tested (Supplementary Figure 1B).”

4. Line 125: “We next examined the binding of bNAbs to infected cells. All antibodies recognized cells infected with the three viral strains, but to different extents”. This validation of binding capacity is an important piece of information and should be presented already with Figure 1.

We have moved the data on the binding capacity of bNAbs to the Figure 1. We agree that this greatly improves the presentation of the results.

5. Fold differences in p24 content and GAG MFI are in some Ab-virus combinations significantly different from untreated cells but the difference is in several cases very modest. In fact only few combinations reach a 50% difference. These are V3 glycan bnAbs against CH058 and vKB18 for both p24 and GAG measurements and V1/V2 bnAbs against vKB18 for p24. The authors need to discuss what impact such a low reduction in infectivity may have.

We fully agree that the decrease in p24 release is sometimes modest. This is now commented in the discussion (lines 231-233):

“However, among the 17 bNAbs and the three strains tested here, only a few combinations of antibodies and viral isolates reached a reduction of p24 release above 50%. It remains to be determined how this may contribute to their overall antiviral efficacy.”

6. The differences between some bnAbs in the retention activity are not easy to comprehend. Why would

10-1074 and PGT128 which are highly related react differently? PGT128 is the best binder to CH058 and AD8 (Figure 3B) but has not effect on p24 release and is not the best in retention (Figure S2)?

We agree with this reviewer that some differences in the capacity of antibodies to trigger viral aggregation are not fully understood. This is commented in the discussion (lines 238-247):

“Unexpectedly, some antibodies that strongly bound to infected cells poorly tethered viral particles. This indicates that binding alone is not sufficient to retain viral particles, and that aggregation of different virions necessitates the attachment of antibodies to selective regions of Env. For instance, PGT128 is unable to trap viral particles, while targeting the same N332 glycan patch as 10-1074. Thus, subtle differences in epitopes influence viral retention by antibodies. Indeed, antibody-mediated tethering by antibodies likely requires efficient binding to Env anchored in both viral and in cellular membranes, as well as the capacity to bridge two virions, which may not be possible for all antibodies or epitopes. It will be important to determine the nature and composition of viral aggregates triggered by bNAbs to determine whether cellular proteins or components of the extracellular matrix contribute to their formation.”

7. In this context, it is often not entirely clear from the text whether an observed effect of an Ab is observed for tested strains or not. Better attention needs to be paid to the differential patterns.

We have modified the text to improve the clarity in multiple instances:

Page 3:

Lines 121-122: “We selected the CH058 strain and 10-1074, 3BNC117, N6, PGT128, PGDM1400 and 10E8 for this analysis.”

Page 4:

Line 129: “We performed a RT-qPCR assay on CH058-infected cells”,

Lines 133-134: “We next determined by confocal microscopy the localization of Gag in CH058-infected cells exposed to 10-1074.”

Line 137: “We performed correlative light and scanning electron microscopy of CH058-infected cells.”

*Lines 151-152: “Similar results were obtained with the AD8 and vKB18 strains (**Supplementary Figure 7**).”*

Lines 158-159: “Coupled to immunogold staining of 10-1074 in CH058-infected cells.”

Lines 165-166: “Similar results were obtained in CH058-infected CHME microglial macrophage-like cells cultured in the presence of 10-1074”

Page 5:

Lines 179-180: “we generated viral aggregates by treating CH058-infected cells with 10-1074 for 24h”

Lines 187-189: “CH058 and vKB18 were less sensitive to neutralization. CH058 was not neutralized by V1/V2-targeting bNAbs (PGDM1400 and PG16) while vKB18 resisted those targeting the MPER and the gp120/gp41 interface”

The correlation analysis in Figure 1F, 3C, and 3D should also be performed separately for the individual viruses, as it seems to be strongly driven by the vKB18 data.

We have performed the correlation analyses separately for each virus (new Supplementary Figures 3 and 11D; see below). All correlations remain significant when considering the three strains separately.

Supplementary Figure 3:

Supplementary Figure 11D:

8. Line 106: “PGDM1400 and 10E8, which did not increase cell-associated Gag by flow cytometry, did not promote p24 accumulation”. This statement is not entirely correct as it only reflects CH058 results. Data for isolates AD8 and vKB18 are different. Importantly, PGDM1400 did increase cell associated Gag for those strains by flow cytometry, Figure 1E).

We agree that the sentence was not entirely correct. We have rephrased the description of the Figure 2 to clarify that only the CH058 strain was used. It now reads (lines 126-128):

“PGDM1400 and 10E8, which did not increase cell-associated Gag by flow cytometry with this strain, did not promote p24 accumulation (Supplementary Figure 5B).”

9. Line 130: “nnAbs and sera from 16 ART treated patients (at a 1:100 dilution) did not reach the levels of binding of bNAbs and were unable to trap HIV-1 at the cell surface.” This is not correct, AD8 binding is good for nnAbs according to Figure 3B.

Indeed, nnAbs retain a certain level of binding against AD8-infected cells. We have thus rephrased the sentence. It reads (lines 110-112):

“Consistently, sera from 16 ART-treated patients (at a 1:100 dilution) did not reach the levels of binding of the most potent bNAbs and were unable to increase Gag levels in CH058-infected cells.”

We have performed an additional analysis, where the MFI of binding of mAbs capable of viral retention was compared to those that do not trap viral particles. These data are presented in a new figure S3C, and commented as follows (lines 109-110):

“Antibodies inhibiting viral egress displayed a higher MFI of binding to infected cells than those that did not (Supplementary Figure 4D).”

10. Several antibodies show enhancing effects in the p24 release data (Figure S2) that are in the extent higher than the inhibitory effects seen with some bnAbs. This is significant of nnAb 4-42 for vKB18. By simple eye-balling it seems surprising that not more of these “enhancing” effects reach significance.

It is indeed true that some nnAbs increase to some extent the levels of released p24. We repeated the statistical analysis, and again, significance was reached only for 4-42 on vKB18, since enhancement was not always reproducible across experiments. Of note, none of antibodies for which enhancement was visually suspected was associated to a parallel decrease in the level of cell-associated Gag. Studying this increase is of interest and would deserve further investigation but is out of scope of the present study.

11. Neutralization data of all antibodies against the probed viruses should be shown and correlated to virus retention capacity.

Neutralization of all mAbs against the three HIV-1 strains tested has been performed and data are now provided in the new Figure 4B. Correlation with retention is now displayed in Figure 4C and Supplementary Figure 11D.

12. The authors should discuss which of the probed Abs have the capacity of bivalent binding of two virions. This will sterically not be possible for all Abs and likely will lend some explanations to the observed patterns.

Structural insights into the recognition of Env by bNAbs help understanding their antiviral functions. In the case of bNAbs, it is well known that steric clashes dictate the number of antibodies that may bind to a single trimer of Env. For instance, CD4bs- and V3-targeting bNAbs bind on the side of Env, and each of the Env protomer can be bound, resulting in three antibodies per Env trimer while V1/V2 targeting antibodies bind on a quaternary structure on the apex, limiting the number of antibodies to 1 or 2 per trimer. However, to our knowledge, how this may influence the capacity of antibodies to bridge virions is not yet described. Such steric limitations will greatly influence aggregation of viral particles. This is now discussed in the manuscript (lines 243-245):

“Indeed, antibody-mediated tethering by antibodies likely requires efficient binding to Env anchored in both viral and in cellular membranes, as well as the capacity to bridge two virions, which may not be possible for all antibodies or epitopes”

13. The discussion is very interesting to read as the authors layout the many open questions that remain to be addressed. Considering that there are many loose ends, short-comings of the current study should be noted. The fact that the impact of virus trapping on virus spread may be limited based on the present results, needs to be discussed.

This is now discussed (lines 231-233):

“However, among the 17 bNAbs and the three strains tested here, only a few combinations of antibodies and viral isolates reached a reduction of p24 release above 50%. It remains to be determined how this may contribute to their overall antiviral efficacy.”

14. “Glycans-V3 loops” is a rather unusual term.

We have changed all “Glycans-V3 loop” and “Glycan-V1/V2 loop” by “V3 loop” and “V1/V2 loop”, respectively.

15. Line 38: Since all other bNAb epitopes are mentioned, silent face bNAbs should be listed as well

Silent face bNAbs are now mentioned (lines 38-41):

“They target conserved sites of vulnerability at the surface of Env: the CD4 binding site (CD4bs), the N-glycans associated to the V1/V2 and V3 loops, the silent face of gp120, the membrane proximal external region (MPER) of gp41 and a larger site spanning the interface between gp41 and gp120.”

16. Line 79: Why were the specific three virus strains chosen? Subtype and sensitivity to the probed Abs and sera should be provided.

We have chosen these three strains to include a lab-adapted strain, a transmitted-founder strain and a primary strain that we isolated from the reservoir of an ART-treated patient. We have determined the sensitivity of these three strains to all the mAbs used in our study (cf. comment n°11). We have also indicated the clade of vKB18 at its first mention (lines 87-89):

“We used three HIV-1 strains: the lab-adapted AD8 isolate, a transmitted/founder strain (CH058) and a clade B virus isolated from the reservoir of an ART-treated patient (vKB18)”

17. Line 167: The authors propose “a Fc-independent non-neutralizing antiviral activity of HIV-1 antibodies”. I am not sure that there is already enough data to state that this capacity is distinct from the neutralization activity. The virions in the aggregates may be neutralized, high affinity (as required for potent neutralization) may be the basis of efficient trapping

We fully agree with the reviewer that aggregation may be a consequence of high affinity and thus neutralization. We have rephrased this sentence to remove the claim on the non-neutralizing nature of this function (lines 223-224):

“Thus, the viral retention by bNAbs reported here may be considered as a Fc-independent antiviral activity of HIV-1 antibodies.”

18. Figure S6B: Please indicate concentration of antibody used.

The concentration of the antibodies was indicated in the legend of Supplementary Figure 7 that now reads:

“All antibodies were used at a concentration of 100 nM.”

Reviewer #2 (Remarks to the Author):

Inhibition of viruses by formation of aggregates is a very classical phenomenon largely describe as the first mechanism participating in virus inhibition. In this manuscript, authors propose that this aggregation occurs by certain bNAbs during virus released on infected cells. This is an interesting additional mechanism of inhibition, but additional experiments should be added in order to proof that indeed, the virus detected is virus produced and released from cells and not the virus inoculum forming aggregates close to cell membrane.

We thank reviewer #2 for his/her evaluation of our manuscript and raising important points about the origin of tethered viral particles.

In the experimental conditions described in this manuscript, author detect very high infection rates: from 2 to 15% infection for primary CD4 T cell (supplemental figure 1). However, in the result section, authors claimed that AZT and 3TC were used to “halt viral replication and avoid confounding effect due to neutralization” (line 82). This sentence is not clear, and misleading especially because later on, authors stipulate, “Ab inhibit virus released from infected cells” (line 23 and lane 160)? It would be extremely relevant to determine whether virus replicate or not under their experimental conditions. If the virus detected by cytometry and by ELISA correspond to virus produced after 24H (AZT and 3TC)+ 24H(AZT and 3TC + Ab) ie 48 H culture, why adding AZT +3TC. What would be this infection rate in the absence of inhibitors AZT+3TC? If AZT and 3TC is added at “suboptimal concentrations”, what is the rational of adding these inhibitors without stopping HIV infection?

We thank reviewer #2 to raise this important point of our experimental protocol. In our experiments, AZT and 3TC are added only after two days of infection, at the time of bNAbs addition. In other words, we infect the cells in the absence of any antiretroviral drug or antibody, wait to obtain 2-15% of the cells infected, and then add antibody and antiretrovirals to study how they impact on viral release from these productively infected cells. AZT and 3TC (reverse-transcriptase inhibitors) act early in the viral cycle and do not impact viral production in cells that are already productively infected.

Since all antibodies do not have a neutralization potential, we think that halting the replication process prior to any evaluation of non-neutralization functions allows for a more accurate comparison.

We have modified the Figure 1A to improve the presentation of our protocol:

We also have modified the text to improve clarity (lines 78-85):

“We infected primary CD4 T cells with HIV-1 for two days and then subjected infected cells to treatment with a panel of bNAbs or an isotype control (mGO53) for 24h. Since bNAbs potentially neutralize HIV-1 spread, we added antiretrovirals (azidothymidine [AZT] and lamivudine [3TC]) during bNAbs treatment, in order to halt viral replication and avoid any confounding effect due to neutralization (Figure 1A). This strategy allowed the normalization of the frequency of infected cells across isotype- and bNAb-treated conditions (Supplementary Figure 1A). AZT and 3TC are reverse-transcriptase inhibitors that act early in the viral cycle, without interfering with p24 production by cells productively infected at the time of addition.”

Finally, we have evaluated how the virus replicates in our experimental conditions and how the antiretroviral treatment impacts spread. We measured the frequency of Gag⁺ infected cells. This data is now presented in the new Supplementary Figure 1A.

On contrary, if inhibitors (AZT and 3TC) were added to “halt viral replication”, the virus detected may correspond to the inoculum that forms immune complexes (virus/Ab) and bind directly on cells without infection (increased MFI of infected cells) and as a result, decreases the residual virus in the supernatant (p24 in ELISA). The conclusion would then be completely different.

We agree that the conclusion would be different in the absence of viral replication. In our case, viral spread is stopped after a few days, to analyze how bNAbs affect viral production by infected cells. Cells were washed prior to addition of the bNAbs and antiretroviral drugs. Thus, cell-free virions from the inoculum are removed prior to the evaluation of aggregates formation. We have modified the Figure 1A to present the entire protocol and improve clarity:

This accumulation of HIV on target cells by Abs may even be detrimental (enhancement at low Ab concentration!) for further HIV replication. Therefore, authors should demonstrate (by kinetic experiments, washing out virus inoculum after infection.... for example) that the virus detected on CD4 T cells effectively correspond to virus released from infected cells and not to residual inoculum trapped on cells via HIV specific Abs.

As explained above, the viral inoculum is washed in all our experiments, suggesting that the virus detected on CD4 T cells does not originate from viral input but from de novo viral production.

The methods section has also been made clearer and now reads (lines 338-340):

“Culture of infected cells with bNAbs

After washing out viral inoculum, HIV-1-infected primary CD4 T cells were cultivated for 24h at 1×10^6 cells/mL in presence of IL-2 (50 IU/mL; R&D Systems), azidothymidine (AZT; 10 nM; Sigma) and lamivudine (3TC; 2.5 nM; NIH AIDS Reagent Program).”

At the concentration used, the new bNAbs had almost no neutralizing activity on primary cells and viruses (see Cohen et al., J. Virol 2018 Feb 12;92(5):e01883-17. doi: 10.1128/JVI.01883-17. Print 2018 Mar 1.). At higher concentrations however, 10-1074 should neutralize infection. Therefore, the increased MFI detected in S2 D is very surprising. Again, what would be the inhibitory activity of 10-1074 at those concentrations in the absence of AZT + 3TC? Authors should analyze the neutralizing profile of the Abs added one hour after infection on primary cells using the same experimental conditions but in absence of AZT+3TC.

We have performed the experiment suggested by reviewer #2 and the results can be seen below. When added 1h post-infection (strain CH058) in the absence of AZT/3TC, the bNAbs recognizing infected cells also neutralized the infection.

The increase in MFI in supplementary Figure S2D is not due to neutralization as it is evaluated in the presence of antiretrovirals (see above and new Figure 1A for a complete overview of the protocol). The increase in MFI is thus promoted by the addition of 10-1074 to infected cells.

Additional comments

Why using the unique concentration of 100nM to compare the different mAbs knowing that the different HIV specific Ab used in this manuscript bind with different affinity on gp120, trimeric spike and viruses. This aggregation activity may be observed for all HIV specific Abs able to recognize HIV if used at optimal binding concentration. Please comment.

Antibodies have indeed different concentrations for optimal binding. This concentration also varies for a given bNAb according to the viral strain tested. Our aim was to provide a proof-of-concept that bNAbs trap viral particle at the plasma membrane. Therefore, we did not adapt the concentration for each antibody again each strain. We cannot exclude that inactive bNAbs in our study would tether viral particles at a higher concentration. This is now discussed in the manuscript (lines 234-236):

“We tested antibodies at 15 µg/mL (100 nM). This is not necessarily the optimal binding concentration which may vary depending on the antibody. We cannot exclude that viral aggregation would be more efficient for some antibodies at higher concentrations.”

What is the rational of using DEA/Dextran. DEA/dextran will modify cell membrane. Does it influence virus capture on CD4 T cells? Does virus/Abs aggregate at the surface of cells without DEA/Dextran?

We have used DEAE/Dextran in order to maximize infection rates, as classically performed, as for instance (PMID: 31833228). However, it is washed together with the viral inoculum prior to addition of bNAbs.

However, as the reviewer suggested, we performed infections (strain CH058) of CD4 T cells in the absence of DEAE/Dextran, washed the inoculum, and let the cells get infected for 48h before washing out the medium and adding bNAbs. We measured retention of viral particles by flow cytometry (left panel) and viral release in the supernatant by a p24 ELISA (right panel). As previously observed with DEAE/Dextran, 10-1074 IgG increased cell-associated Gag levels and decreased viral release, whereas 10-1074 Fab did not. Thus DEAE/Dextran does not affect the ability of 10-1074 to inhibit viral release.

We have added a sentence in the methods (lines 313-315):

“Of note, viral release was inhibited by 10-1074 irrespectively of the presence of DEAE/Dextran during the initial infection of T cells.”

REVIEWER COMMENTS

Reviewer #1 (Remarks to the Author):

Thank you for the thorough reply. All my queries were addressed.

Reviewer #2 (Remarks to the Author):

In this manuscript, Dufloo et al nicely show virus retention at the surface of infected cells and added interesting additional data to prove this phenomenon. However, they are still experiments missing to demonstrate that this increased amount of virus particles aggregated with bNAbs are formed flowing virus budding.

The supplemental figure 1A added in the new version show that the percentage of infected cells drop at day 3. This drop was not due to AZT/3TC treatment, nor to bNAb treatment. Overall, addition of AZT/3TC at day 2 did not modify the kinetic of % infected cells at day 3. Therefore, the sentence “we added antivirals AZT and 3TC during bNAbs treatment in order to halt viral replication” is not accurate. The AZT/3TC treatment of the cells need to be reconsidered in line of results shown figure S1A. Note that the AZT/3TC concentration used is quite low.

According to supplemental figure 1A, there is no data suggesting that new virus production occurs at day 3. On contrary, % infected cells strongly decreased. What is the reason for this drop of infected cells (lysis of infected cells?). The strong decrease of infected cells at day 3, one day after the washing step should be comment. Also the high potency of neutralizing activity of bNAbs shown in the figure 4B is not in line with S1A figure showing that bNAb 10-1074 do not modify the % of infected cells at day 3 and only slightly affect the percentage of infected cells at day 4 and 5.

The washing step now added figure 1A of the new version is a crucial step for the analysis of virus budding and release. This washing procedure should be better explain. Indeed, extensive washing should be performed to get read of virus particles trapped at the surface of infected cells. Indeed, virus is efficiently trapped by alternative receptors and bNAbs may form aggregates and stabilized free virus produced day 2 and trapped close to the membrane. This phenomenon is certainly of interest, but in this case, the mechanism would be general mechanism of aggregation of free virus particles and retention at the surface of cells. This mechanism may also be of interest to decrease virus infectivity.

In order to demonstrate that bNAbs retain virus produced at the surface of infected cells (viral retention on infected budding cells), authors should demonstrate that the virus detected on infected cells (24 hours after treatment of infected cells) is effectively virus produced and not residual virus

trapped at the surface of infected cells. Authors should show that the washing procedure performed has eliminated the residual virus trapped on infected cells. A kinetic experiment with electron microscopy performed at different time points after the washing step would allow unraveling this point by following new virus production over time.

Additional comment: Concentration used for Ab (whole IgG and Fab) should be mentioned figure 4E. Please note that the Fab fragment has a 3-fold decrease molecular weight compared to IgG. Therefore, molarities should be used and not $\mu\text{g/ml}$ for comparison.

Reviewer #1 (Remarks to the Author):

Thank you for the thorough reply. All my queries were addressed.

We thank reviewer #1 for his/her positive feedback

Reviewer #2 (Remarks to the Author):

In this manuscript, Dufloo et al nicely show virus retention at the surface of infected cells and added interesting additional data to prove this phenomenon. However, they are still experiments missing to demonstrate that this increased amount of virus particles aggregated with bNAbs are formed following virus budding.

We thank reviewer #2 for his/her evaluation of our study.

We agree that our data did not formally demonstrate that viral aggregates are formed from budded viral particles, although this was the most likely hypothesis. Thus, we have added new data supporting this hypothesis:

- We now show that infected cells are producing viral particles during our period of analysis, even when the peak of viral replication is reached.
- We demonstrate that infected cells subjected to extended washing display the same phenotype of viral aggregation.
- We analyzed bystander non-infected cells and show that they are not covered by aggregates, while productively infected cells are.
- We performed a time course analysis by flow cytometry to show that aggregates are not detected at very early time points and that they appear overtime at the surface of infected cells.
- We provide new images of transmission electron microscopy and immunogold labeling that clearly show that viral aggregation occurs during budding of viral particles, that are tethered with mature virions
- We discuss previous work from our lab (including a J. Exp Med 2013 and JVI 2017) showing that the incubation of bNAbs with productively infected cells inhibits cell-to-cell viral transmission and impairs the capture of cell-free viruses by target cells, supporting the hypothesis that aggregates form from infected cells rather than they are captured by them.

Finally, we moderated our message and now discuss the possibility that previously budded particles could also be incorporated into aggregates during their formation.

The supplemental figure 1A added in the new version show that the percentage of infected cells drop at day 3. This drop was not due to AZT/3TC treatment, nor to bNAb treatment. Overall, addition of AZT/3TC at day 2 did not modify the kinetic of % infected cells at day 3. Therefore, the sentence “we added antivirals AZT and 3TC during bNAbs treatment in order to halt viral replication” is not accurate. The AZT/3TC treatment of the cells need to be reconsidered in line of results shown figure S1A. Note that the AZT/3TC concentration used is quite low.

We agree with the reviewer that the difference observed at day three with or without AZT/3TC is minor. However, as depicted below for the reviewer, this difference is statistically significant, at all days following treatment, including day 3. This experiment demonstrates that AZT/3TC treatment is effective in our settings. We thus believe that our description of the results is

accurate. We have added this panel and the statistics has been included to the existing figure 1A.

Figure: Mock and AZT/3TC conditions were compared using a Wilcoxon non-parametric test. n=6 donors of CD4 T cells.

The reviewer is right that 10 nM of AZT and 2.5 nM of 3TC are low concentrations. This was a mistake. The molecules were used are 10 μ M and 2.5 μ M. We apologize for the error, which has been corrected.

According to supplemental figure 1A, there is no data suggesting that new virus production occurs at day 3. On contrary, % infected cells strongly decreased. What is the reason for this drop of infected cells (lysis of infected cells?). The strong decrease of infected cells at day 3, one day after the washing step should be comment. Also the high potency of neutralizing activity of bNAbs shown in the figure 4B is not in line with S1A figure showing that bNAb 10-1074 do not modify the % of infected cells at day 3 and only slightly affect the percentage of infected cells at day 4 and 5.

We agree with the reviewer that Figure 1A shows that peak viral replication is reached at day 3. However, this figure shows the frequency of infected cells, providing a snapshot of the number of infected cells, rather than viral production per se. Indeed, viral production is rather stopped when no more infected cells are detectable, something that doesn't not occur within the time frame of our analysis.

To confirm that viral production occurs, we quantified p24 in the supernatant, after addition of AZT/3TC. As visible below, p24 increased overtime even if peak replication was reached and further infection of new targets was blocked by AZT/3TC. This demonstrates that the infected cells are producing p24 in our conditions, even if their frequency is decreasing, most likely because of viral cytopathic effect. This result has been added as a new supplementary Figure 1B.

The result section has been appended accordingly (page 3):

"p24 production is reduced but not halted by addition of AZT-3TC (Supplementary Figure 1B)."

Viral production was further confirmed by the observation of budding sites by transmission electron microscopy in cells treated with a isotype control antibody (24h after addition of AZT/3TC and 3 days after infection) (Supplementary figure 8A):

Red arrows indicate budding sites.

As the drop in infected cells might be misleading, we have included a description of this results in the method section. It reads (page 3):

"To further avoid any confounding effect of viral replication, we treated cells with antibodies at the time of peak viral replication (Supplementary Figure 1A)."

The dynamics of viral replication in our system also helps explaining why bNAb may appear less effective as compared to a classical neutralization assay performed in Figure 4B. Indeed, bNAb were added at the time of peak viral replication. Their action is to inhibit infection of novel cells, a process which is already decreasing at this time. As seen below, bNAb significantly decreased viral replication after their addition to the culture (at day 2). We added these statistics to Supplementary Figure 1A.

Figure: Isotype and 10-1074 (bNAb) treatment in the absence of AZT/3TC were compared using a Wilcoxon non-parametric test. n=3 independent experiments (6 donors of CD4 T cells).

The washing step now added figure 1A of the new version is a crucial step for the analysis of virus budding and release. This washing procedure should be better explain. Indeed, extensive washing should be performed to get read of virus particles trapped at the surface of infected cells. Indeed, virus is efficiently trapped by alternative receptors and bNAbs may form aggregates and stabilized free virus produced day 2 and trapped close to the membrane. This phenomenon is certainly of interest, but in this case, the mechanism would be general mechanism of aggregation of free virus particles and retention at the surface of cells. This mechanism may also be of interest to decrease virus infectivity.

We agree with the reviewer that the washing step is important. We have modified the presentation of the protocol to better explain that the medium has been replaced by fresh medium. It reads (page 3):

“To this aim, we infected primary CD4 T cells with HIV-1 for two days, washed the cells to replace the medium and then subjected infected cells to treatment with a panel of bNAbs or an isotype control (mGO53) for 24h”

Furthermore, we performed a novel experiment to assess how the washing may influence aggregation. We compared our standard protocol of washing to (i) an extended washing process (3 PBS washes) and (ii) an extended washing and a proteolytic digestion of cell surface proteins (3 PBS washes followed by 30 min of trypsin treatment). Of note, we validated the efficacy of the trypsin treatment by measuring CD4 levels at the end of the treatment. As depicted below, aggregation occurs regardless of the washing protocol used.

These data have been added to a new Supplementary Figure 2A and the results section has been modified accordingly (page 3):

“Adding extra PBS washes or a proteolytic degradation of cell surface proteins by trypsin prior to bNAbs treatment has no impact on the increase in MFI induced by 10-1074 (Supplementary Figure 2A).”

The hypothesis that *“virus is efficiently trapped by alternative receptors and bNAbs may form aggregates and stabilized free virus produced day 2 and trapped close to the membrane”* is interesting. Such an infection-independent phenomenon would imply detection of aggregates on both infected cells and bystander non-infected cells. Thus, we performed new experiments to analyze bystander cells, to assess the formation of viral aggregates at their surface after washing and replacing the medium to get rid of existing viral particles. As shown in the new Supplementary Figure 5E, we did not detect aggregates on non-productively infected cells, while aggregates are observed on productively infected cells. These results confirm our previously published observation that bNAbs inhibit viral cell-to-cell transmission by impairing

viral capture by target cells (Malbec et al, J Exp Med 2013 doi: 10.1084/jem.20131244 ; Bruel et al, JVI 2017 ; doi: 10.1128/JVI.02440-16)

New Supplementary Figure 5F:

The result section has been appended to include a description of these data and to discuss our previous work (page 4):

“We next determined by confocal microscopy the localization of Gag in CH058-infected cells exposed to 10-1074. Analyses of 803 cells across 5 donors of CD4 T cells revealed that this bNAb induced a strong accumulation of p17 at the plasma membrane (4.2-fold increase compared to mGO53) (Figure 2C-D and Supplementary Figure 5D-E). As increased p17 levels may be due to capture of extracellular virions, we also analyzed 150 non-infected bystander cells from these 5 donors. This analysis reveals a lack of increase of p17 on bystander cells (Supplementary Figure 5F), showing that p17 accumulation by 10-1074 requires productive infection.”

And in the discussion (page 7):

“We did not observe aggregates on bystander non-infected cells, confirming that productive infection is needed to form aggregates. We previously reported that bNAbs decrease viral capture by CD4 T target cells^{36,42}. The mature viral particles present within aggregates may still have been trapped immediately after budding and maturation, or may also represent released virions that have been captured by these large structures.”

Furthermore, Figure 3B clearly shows active viral budding at the sites of viral aggregation. Budding viral particles are in the close vicinity of released-but-trapped virions. We have modified Figure 3 to highlight with a yellow arrowhead novel budding sites, and included a description of this observation in the results section.

New figure 3B:

We also added novel images of active viral budding at site of aggregation in the new Supplementary Figure 10:

Supplementary Figure 10 (related to Figure 3): Infected CD4 T cells (CH058) cultivated for 24h with a bNAbs (10-1074) were stained with an anti-human IgG antibody coupled to colloidal gold beads and analyzed by transmission electron microscopy (TEM). Magnifications of areas with viral accumulation at budding sites are shown. Scale bar: 200nm-500nm. The cell donor and the cell ID is indicated for each image. Red arrowheads point colloidal gold beads, indicative of 10-1074. Yellow arrowheads indicate budding virions.

The results section now reads:

“Trapped virions make contacts with budding viral particles (Figure 3B and Supplementary Figures 10)”.

We also added an image of TEM showing bNAbs immunogold in between a viral particle and a budding virion. It has been included in the Figure 3 as a new panel D:

We included a description of this image in the result section:

“We further observed a bNAb immunogold staining in between a budding virus and a tethered virion (Figure 3D)”

Altogether, these data strongly suggest that aggregation occurs in close proximity of active viral budding sites. However, as we cannot formally rule out that already budded viral particles are also incorporated within aggregates, we discuss this possibility, and the text modifications include:

Abstract:

“We combined immunofluorescence, scanning electron microscopy, transmission electron microscopy and immunogold staining to reveal that some bNAbs form large aggregates of virions at the surface of infected cells.”

Discussion page 7:

“Our data strongly suggest that bNAbs retained virions at budding sites. The sites of aggregation observed by transmission electron microscopy contained budding viral particles. We did not observe aggregates on bystander non-infected cells, confirming that productive infection is needed to form aggregates. We have previously reported that bNAbs decrease viral capture on CD4 T target cells^{36,42}. The mature viral particles present within aggregates may still have been trapped immediately after budding and maturation, or may also represent released virions that have been captured by these large structures.”

In order to demonstrate that bNAbs retain virus produced at the surface of infected cells (viral retention on infected budding cells), authors should demonstrate that the virus detected on infected cells (24 hours after treatment of infected cells) is effectively virus produced and not residual virus trapped at the surface of infected cells. Authors should show that the washing procedure performed has eliminated the residual virus trapped on infected cells. A kinetic experiment with electron microscopy performed at different time points after the washing step would allow unraveling this point by following new virus production over time.

The several lines of evidence provided above demonstrate that viral aggregates contain newly produced viral particles. It is now discussed that already budded virions may also be incorporated within aggregates. We believe that the EM images that we have included are of sufficient quality to support our conclusions. Thus, instead of performing a novel series of electron microscopy at different time-points, we did a kinetic analysis by flow cytometry on two donors of CD4 T cells infected with CH058 and treated with 10-1074. It shows that viral aggregates are not present at the surface of infected cells at t=0h, but form overtime.

This panel has been included in the Supplementary Figure 1. The results section now reads (page 3):

*“A kinetic analysis performed with 10-1074 and CH058-infected cells revealed that Gag MFI increases over 24h (**Supplementary Figure 1E**). »*

Additional comment: Concentration used for Ab (whole IgG and Fab) should be mentioned figure 4E. Please note that the Fab fragment has a 3-fold decrease molecular weight compared to IgG. Therefore, molarities should be used and not $\mu\text{g/ml}$ for comparison.

We agree with the reviewer that Fab fragment has a lower molecular weight than full IgG, and thus required to be compared at similar molarity rather than concentration. Antibodies, and their various fragments have thus been tested at 100 nM, as stated in the methods. For clarity we have indicated the tested concentration in the legends and in the text when Fab and F(ab')_2 are compared.

Results now read (page 4): *“Antibodies and their fragments were tested at 100 nM.”*

REVIEWERS' COMMENTS

Reviewer #2 (Remarks to the Author):

Authors respond to my queries and corrected the mistakes raised

REVIEWERS' COMMENTS

Reviewer #2 (Remarks to the Author):

Authors respond to my queries and corrected the mistakes raised

We would like to thank reviewer #2 for his/her positive feedback and his/her help to improve our manuscript.